# Intrinsically flexible multimode reconfigurable transistors for polymorphic circuits and neuromorphic devices

Wanting Wang[1,2], Rui Qiu[2], Jiahao Zhu[2], Tianyu Zhu[2], Jialiang Wang[3], Dexing Liu [1], Jiaqiao Liang[1], Chunxiu Wang[1,2], Sixin Zhang[2], Zifan Wang[2], Qiuyue Huang[2], Xinwei Wang [3] & Min Zhang [1,2] ✉

Reconfigurable transistors hold considerable significance for both integrated circuits and neuromorphic electronics. At the same time, flexible electronics is developing rapidly, promoted by various emerging applications, such as wearable electronics and smart robots. Therefore, flexible reconfigurable transistors are highly expected to create emerging application scenarios while they are seldom reported. Here, intrinsically flexible multimode reconfigurable transistors (IFMRTs) with dual-gate structure are proposed and realized. Either top or bottom gate can serve as the modulation terminal to switch the transistor among p-type, n-type, and ambipolar modes. The threshold voltage of the reconfigurable transistor is modulable. Based on IFMRTs, we have demonstrated inverters and polymorphic logic circuits with key bit-selectable NAND or NOR logic functions, which provides solutions for hardware security. We have also demonstrated artificial heterosynapse and dendrite, achieving reconfigurable synaptic responses and dendrite integration. Taking the simulation of automatic obstacle avoidance and coordination control of hand movement by low-level and high-level neural centers as examples, we illustrate the potential intelligence of IFMRTs in robotic decision-making and arm control. IFMRTs maintain their reconfigurability even after 5000 bending cycles at 4 mm radius. The design and demonstration of IFMRTs open up possibilities for flexible wearable electronics and soft intelligent robots.

As silicon transistors approach the physical limits of technology scaling, the progression of Moore's Law has encountered a bottleneck[1]. On the other hand, the insatiable demand for enhanced computational capacity to process and analyze voluminous data continues to drive research into novel types of devices. Among the emerging technologies, reconfigurable transistors have opened up a promising path at the device level, enabling reconfigurability between p-type and n-type transistors by simply changing the electrical input[2–5]. The operational flexibility of reconfigurable transistors facilitates the miniaturization and multifunctionality of electronic products, enriching the expressive

capability at circuit level[6–11]. Furthermore, reconfigurability provides a solution for hardware security since the reconfigurable designs can resist easy reverse deciphering[12,13]. Reconfigurable transistors represent a pivotal advancement, accelerating breakthroughs in the field of integrated circuits.

At the same time, brain-like intelligence provides an effective method of information processing, which aims to emulate the efficient computing neural network in the human brain, with neurons as fundamental units connected to each other through synapses and dendrites[14,15]. Artificial synapses and dendrites, capable of emulating

[1]School of Science and Engineering, The Chinese University of Hong Kong, Shenzhen, China. [2]School of Electronic and Computer Engineering, Peking University, Shenzhen, China. [3]School of Advanced Materials, Peking University, Shenzhen, China. ✉e-mail: mzhang@cuhk.edu.cn

neural functions, are crucial for realizing neuromorphic electronics. Firstly, for synapses, most current studies of synaptic transistors have focused on elementary excitatory and inhibitory responses in homosynapses, which are characterized by simple pre- and post-synaptic interconnections[16–18]. Heterosynapses, which have a multi-terminal configuration with two or more pre-synaptic membranes and one postsynaptic membrane, also constitute fundamental building blocks of neural networks[19,20]. One of the critical biological mechanisms of heterosynapses is heterosynaptic plasticity. That is, one of the two presynaptic membranes acts as a modulatory terminal, which can modulate and reconfigure the response between the other presynaptic membrane and postsynaptic membrane[21–26]. For instance, endorphins can act as neuromodulators to alleviate subjective pain caused by noxious stimuli for humans[24]. Besides, the co-release of excitatory and inhibitory neurotransmitters from a single axon terminal in neurons of the ventral tegmental area exemplifies the reconfigurable nature of synaptic responses between excitatory and inhibitory behaviors[22]. This organism-intrinsic synaptic mechanism turns out to be indispensable and plays a pivotal role in resolving the interplay between local and global neuro-modulations. It enhances the freedom of neural regulation, contributing significantly to maintaining the steady-state of the nervous system[19]. Furthermore, for dendrites, the branches of a postsynaptic neuron to connect to other multiple presynaptic neurons, are critical components of neural information integration[27,28]. The inputs from presynaptic neurons are delivered to presynaptic membranes. The postsynaptic dendritic membrane gathers and preprocesses the inputs and subsequently transmits the signal to the next neuron. The information reception and processing capability of dendrites at the single neuron level, such as logic operations and integration with filtering, stands as a fundamentally essential and effective mechanism within neural network[29–31]. Mimicking dendrite integration by multi-terminal electronic devices is highly desired. In brief, reconfigurable transistors as neuromorphic devices are expected to achieve synaptic and dendritic functions for the development of advanced neuromorphic electronics.

Meanwhile, with the emergence of wearable devices[32,33], smart robotics[34,35] and human-machine interfaces[36,37], flexible circuits and neuromorphic devices have been reported to meet the requirement of the conformal fit on human body and robots[16,38,39]. The wearable system integration has further intensified the demand for a multifunctional device platform with the demonstration capability of both circuits and artificial synapses[37]. This triggers the research on developing flexible reconfigurable transistors, which are seldom reported before.

Herein, we have proposed and realized intrinsically flexible multimode reconfigurable transistors (IFMRTs) and demonstrated their applications in circuits, artificial synapses, and dendrite integration. The transistor has a dual-gate structure with a top gate (TG) and a bottom gate (BG), and an all-carbon-nanotube frame, including carbon-nanotube channel and carbon-nanotube source/drain/gate electrodes. To realize reconfigurability, one gate acts as a polarity gate (PG) to modulate the operating mode, and the other gate acts as a control gate (CG) for the channel switching between on and off. Either TG or BG can be used as PG, while TG exhibits stronger channel coupling and modulating capability than BG does. The ambipolarity of the channel and the modulation capability of the PG enable the reconfigurability of the transistor among p-type, n-type, and ambipolar modes. Furthermore, the threshold voltage can also be modulated accordingly in either p- and n-type mode. At the same time, the great intrinsic flexibility of the devices enables the reconfigurable operation under bending condition, and the reconfigurability can be maintained even after 5000 times bending at a radius of 4 mm. Furthermore, we implement inverters and polymorphic logic circuits based on the p- and n-type IFMRTs. Two operating modes of one circuit, NAND and NOR, are achieved through a selection of a key bit. This design can hide

the actual circuit function, making it ideal for applications in hardware security. We also demonstrate artificial heterosynapses to emulate heterosynaptic plasticity. Besides the fundamental synaptic behaviors based on polarization of dipoles in dielectric are exhibited, the synaptic plasticity of this device can be modulated by BG voltage and reconfigured between excitatory and inhibitory modes. For the same presynaptic pulse applied on TG, the synaptic response can be altered from negative to positive or vice versa by setting different BG modulation voltage. Furthermore, by utilizing the different coupling ability of the two gates to the channel, dendrite integration is achieved. The four cases of two-bit binary input signals can be completely distinguished, providing solution for robotic applications such as automatic obstacle avoidance and other similar decision-making scenarios. In addition, we also emulate the mutual coordination of human arm motion control between high-level and low-level nerve centrals, brain and spinal cord, which is expected to facilitate the robotic arm control and intelligent prosthesis. In contrast to previous studies, this work realizes flexible reconfigurable transistors with functional validation in integrated circuits, artificial synapses, and dendritic integration, which significantly advancing their versatility and applicability. By bridging device innovation with function implementations, this work not only expands the potential applications of flexible reconfigurable transistors but also opens pathways for next-generation adaptive electronics and neuromorphic computing (Supplementary Table 1 and Supplementary Note 1).

## Results

### Device structure and characteristics of the IFMRT

Figure 1a presents the schematical diagram of the proposed IFMRT. The two gate electrodes and dielectric layers are positioned at the top and bottom. The optical microscope image of IFMRT is depicted in Fig. 1b. Among the emerging materials, carbon nanotubes (CNTs) have shown excellent electrical and mechanical properties[22]. The CNT thin film also exhibits high electrostatic coupling with the gate electrode, which is advantageous for achieving reconfigurability[2]. In IFMRTs, the source/drain/gate electrodes and the channel of IFMRT are fabricated by metallic CNTs (M-CNTs) and semiconducting CNTs (S-CNTs), respectively, to achieve intrinsically flexible ability. The scanning electron microscopy (SEM) images display the morphology of M-CNT and S-CNT networks, as shown in Fig. 1c, d. With the dual-gate structure of IFMRTs, we adopt an intrinsically flexible hybrid layer design of polyimide and $Al_2O_3$ (hybrid $PI$-$Al_2O_3$) for the dielectrics, which was proposed in our previous work[40]. Molecular layer deposition (MLD) and atomic layer deposition (ALD) are alternatively executed to grow the polyimide and $Al_2O_3$ layers, respectively, with subnanometer thickness, and thus a uniform mixing of the polymer and oxide could be achieved at a nanometer level. This combines the merits of both organic and inorganic materials, endowing the hybrid polyimide-$Al_2O_3$ dielectric with simultaneous intrinsic flexibility and high-κ dielectric performance. Both the all-CNT frame and the hybrid dielectrics contribute to the flexibility of dual-gate IFMRT. Detailed fabrication process of IFMRTs is provided in Methods (Supplementary Fig. 1). The fabricated IFMRT array, as presented in Fig. 1e, f, illustrates the centimeter-level manufacturing capability and micrometer-scale feature size of this platform, as well as the conformality of the IFMRT array to the human finger. The IFMRT array exhibits good transmittance for visible light (Supplementary Fig. 2, Supplementary Table 2, and Supplementary Note 2). Therefore, the all-carbon-nanotube IFMRTs have promising prospects in transparent flexible electronics and wearable systems.

The achieved IFMRT possesses a multimode and multiple-threshold-voltage reconfigurability, as schematically depicted in Fig. 2a. The transistor exhibits the capability to be reconfigured between p-type, ambipolar, and n-type modes, which is attributed to the material design and the structure design. The ambipolarity of the

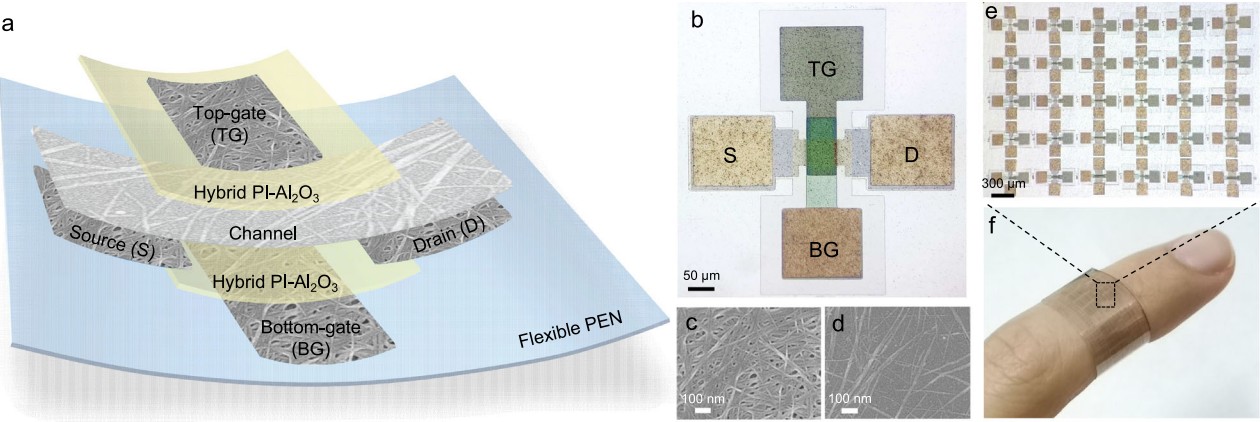

**Fig. 1 | Schematic diagram and characterizations of the proposed IFMRT.**
**a, b** Schematic diagram (**a**) and optical microscope image (**b**) of the IFMRT. **c, d** SEM images of M-CNT network for electrodes (**c**) and S-CNT network for channel (**d**). **e** Optical microscope image of IFMRT array with independent devices. **f** Photograph of the flexible and transparent IFMRT array wound around a human finger.

S-CNT channel arises from the n-type doping by $Al_2O_3$ in the dielectric layer. It can be attributed to the desorption of water and oxygen molecules by the high-temperature and high-vacuum environment during ALD as well as the n-type electrostatic doping by positively charged oxygen vacancies (Supplementary Fig. 3 and Supplementary Note 3). Simultaneously, the dual-gate structure provides the transistor an extra modulation terminal to realize the reconfiguration of different modes by applying different voltages. When TG acts as PG to modulate the polarity of the transistor, BG acts as CG to control the switching on and off. Conversely, when BG acts as PG, TG acts as CG. The reconfigurability of IFMRT in both cases are shown in the Fig. 2b, c, respectively. In the case of TG as PG in Fig. 2b, when the TG voltage ($V_{tg}$) is set to 0 V, the transfer characteristic of the BG transistor in IFMRT is p-type and the on/off ratio stands at $1.73 \times 10^4$. At the same time, there exists obvious ambipolar current on the n-branch. When $V_{tg}$ is adjusted to −0.5 V, the transfer curve shifts right. Inversely, applying a positive voltage to TG, incrementally increasing from 0 V to 3.5 V, results in a continuous leftward shift of the transfer curve. Gradually, the IFMRT is reconfigured from p-type to ambipolar mode and ultimately to n-type mode. The on/off ratio of the n-type transistor also reaches $10^4$. In Fig. 2c, the modulation and reconfiguration capabilities of BG are similar to those of TG. The threshold voltage ($V_{th}$) of p- and n-type modes in both cases is extracted from Fig. 2b, c, and the relationship between $V_{th}$ and PG voltage is shown in Fig. 2d, e. It is evident that the IFMRT has free reconfigurability between p- and n-type polarities and the $V_{th}$ of both p- and n-type transistors are adjustable across high, medium, and low ranges. Either BG or TG can serve as PG to achieve the free modulation.

However, the modulation capability of TG is slightly stronger than that of BG, as illustrated by different $\Delta V_{th}$ in the two cases with the same PG voltage change. In Fig. 2d, in the case of TG acting as PG, when $V_{tg}$ increases from 0 V to 0.5 V and from 3.0 V to 3.5 V, $V_{th}$ changes by 0.44 V and 0.64 V, respectively. But in Fig. 2e, in the case of BG acting as PG, when the voltage of BG ($V_{bg}$) has the same increase, $V_{th}$ changes by 0.32 V and 0.30 V, respectively. The slight difference in curve density observed in Fig. 2b, c further reflects the stronger modulation capability of TG. This characteristic is attributed to the different channel coupling of BG and TG, as schematically depicted in Fig. 2f. The transfer characteristics when sweeping $V_{tg}$ and $V_{bg}$ separately are measured (Supplementary Fig. 4). The smaller subthreshold swing (SS) under TG sweeping illustrates the stronger coupling capability of TG, which is determined by the structure design and fabrication process. The oxygen plasma etching during the fabrication of S/D electrodes has introduced interface states at the surface of the bottom dielectric

layer, leading to a larger SS under BG sweeping and a weaker coupling capability of BG compared with the TG. (Supplementary Note 4). The differentiation of the two gates holds significant implications for artificial heterosynapses, which will be discussed in detail later. Potential solutions to improve the bottom surface and mitigate this asymmetry include scaling down the bottom dielectric thickness to increase the gate dielectric capacitance and decrease SS. Besides, adjusting the oxygen plasma power and exposure duration during M-CNT etching could minimize interface state density in bottom dielectric and enhancing overall device performance.

To theoretically elucidate the modulatory and reconfigurable characteristics of IFMRTs, as manifested in both cases presented in Fig. 2b–c, we extract the channel conductance at CG 0 V under different PG voltages and present the relationship in Fig. 2g, h. Whether BG or TG acts as PG, the conductance exhibits a consistent changing trend with PG voltage, initially decreasing and then increasing. This indicates that PG voltage controls the number and type of carriers initially accumulated in the channel at a CG voltage of 0 V. Figure 2i provides a detailed analysis by the example of the band diagram when BG acts as PG. Firstly, when PG voltage is 0 V, the initial channel conductance is about $10^{-9}$ S and the IFMRT exhibits p-type characteristics. Accordingly, within the energy band diagram, the fermi level is situated slightly below the intrinsic level and there is no highly conductive channel. Upon the application of positive or negative bias to PG, the surface energy band of the channel bends downward or upward, accumulating electrons or holes and forming n-type or p-type channels. The greater the bias applied to PG, the more curved the surface energy band, which responds to the threshold voltage modulation. When a certain PG voltage brings the surface Fermi level proximate to the intrinsic level, the IFMRT approaches the ambipolar mode. Therefore, the modulating effect of PG on the surface energy band and the initial channel conductance elucidates the multimode and multiple-threshold-voltage reconfigurability of IFMRTs. As shown in Fig. 2j, k, the gate leakage current remains below 1 nA regardless of whether TG or BG acting as PG. The output characteristics of p-type and n-type IFMRTs are also measured (Supplementary Fig. 5). The saturation current, output resistance, and intrinsic gain are extracted (Supplementary Note 5). Figure 2l provides the hysteresis curves when BG or TG are scanned. The hysteresis windows are not fixed but can be designed and tuned for a specific application by modulating the MLD and ALD cycles during the dielectric fabrication[41]. That is, the PI-$Al_2O_3$ dielectric can be designed and fabricated to eliminate the hysteresis for logic circuits, while it can be engineered to achieve a significant hysteresis window for synaptic applications.

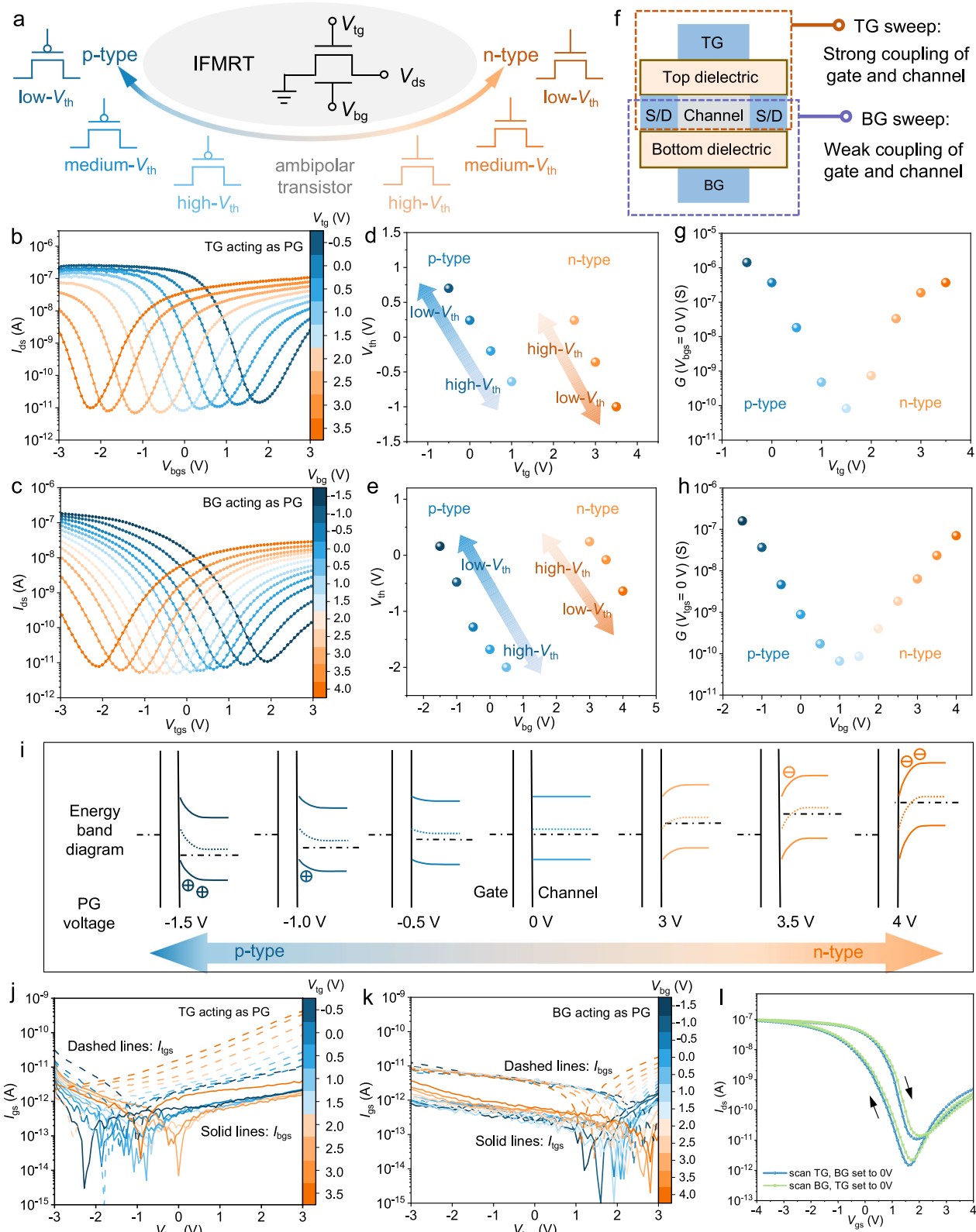

**Fig. 2 | Multimode and multiple-threshold-voltage reconfigurability of IFMRTs.** **a** Schematic illustration of the reconfigurability of IFMRT. **b**, **c** Transfer characteristics of the IFMRT under different PG voltages when TG (**b**) or BG (**c**) acting as PG ($V_{ds} = -0.1\,V$, $W = 50\,\mu m$, $L = 50\,\mu m$). **d**, **e** The $V_{th}$ of the IFMRT in p-type and n-type modes under different PG voltages when TG (**d**) or BG (**e**) acting as PG. **f** Schematic illustration of the different channel coupling capabilities of TG and BG. **g**, **h** The channel conductance of the IFMRT while CG voltage is 0 V under different PG voltages when TG (**g**) or BG (**h**) acting as PG. **i** The energy band diagrams in multiple-threshold-voltage p-type and n-type modes under different PG voltages when BG acting as PG. **j** Gate leakage current when TG acting as PG, corresponding to this figure (**b**). **k** Gate leakage current when BG acting as PG, corresponding to this figure (**c**). **l**, Hysteresis curve of a IFMRT when TG or BG is scanned. ($V_{ds} = -0.1\,V$, $W = 50\,\mu m$, $L = 50\,\mu m$).

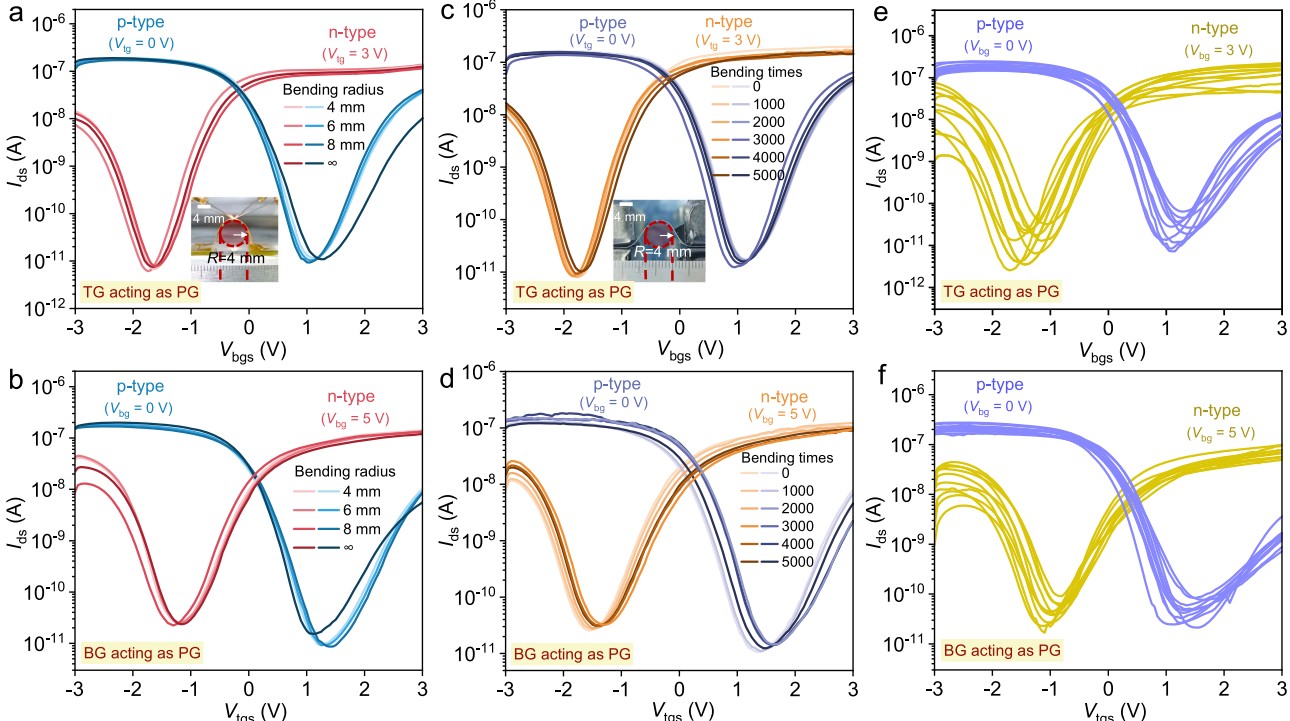

**Fig. 3 | Intrinsic flexibility and statistical data of IFMRTs. a, b** Transfer characteristics of a IFMRT measured in situ at 4, 6, 8 mm and ∞ (flat state) bending radius in p-type and n-type modes when TG (**a**) and BG (**b**) acting as PG ($V_{ds} = -0.1$ V, $W = 50$ μm, $L = 50$ μm); Inset, the photograph of the in situ measurement at 4 mm bending radius. **c, d** Transfer characteristics of the IFMRT measured before and after different bending times at bending radius of 4 mm in p-type and n-type modes when TG (**c**) and BG (**d**) acting as PG ($V_{ds} = -0.1$ V, $W = 50$ μm, $L = 50$ μm); Inset, the photograph of the bending operation by auto-bending instrument. **e, f** Transfer characteristics of 10 IFMRTs in p-type and n-type modes when TG (**e**) and BG (**f**) acting as PG ($V_{ds} = -0.1$ V, $W = 50$ μm, $L = 50$ μm).

The mechanical flexibility of IFMRT is quantitatively evaluated by bending radius and cyclic durability. We conducted in situ measurements of the transfer characteristics for both p-type and n-type IFMRTs, under a flat state and bending radius of 4, 6, and 8 mm, with the TG and BG alternately serving as PG, as shown in Fig. 3a,b. The gate leakage current remains under 100 pA, and the extracted device parameters, $V_{th}$, $SS$, effective mobility, on-current, and off-current, exhibit no obvious degradation (Supplementary Figs. 6 and 7). It indicates that IFMRTs can maintain the mechanical flexibility down to a bending radius of 4 mm, corresponding to a calculated tensile strain of 1.6% (Supplementary Fig. 8 and Supplementary Note 6). The durability tests are shown in Fig. 3c, d. The gate leakage current and extracted parameters are also analyzed (Supplementary Figs. 9 and 10). These further confirm the capability of IFMRTs to withstand the bending cycles of up to 5000 times at a radius of 4 mm. The excellent mechanical flexibility is attributed to the complete adoption of intrinsically flexible materials, CNTs, and hybrid PI-$Al_2O_3$. Moreover, the transfer characteristics of 10 IFMRTs operating in p-type and n-type modes are presented in Fig. 3e, f when TG or BG acts as PG, respectively, with the device parameters extracted (Supplementary Figs. 11 and 12 and Supplementary Note 7). The observed effective mobility asymmetry can be attributed to different Schottky barrier heights for electrons and holes at the metal-semiconductor contact (Supplementary Note 8). The statistical data for devices with three different channel sizes under four operation modes (p-type modulated by BG, n-type modulated by BG, p-type modulated by TG, and n-type modulated by TG) are collected (Supplementary Figs. 13–16). The parameters show a tight distribution. Especially, the average $I_{on}$ follows the expected W/L scaling relationship well. It reflects the reproducibility of IFMRT and its integration potential for high-complexity reconfigurable circuit functions. Besides, adjusting the number of

S-CNTs in the channel can shift the curve and $V_{th}$ of n-type and p-type modes for further improving circuit performance (Supplementary Fig. 17 and Supplementary Note 9).

## Reconfigurable polymorphic logic gates by IFMRT

Based on n-type and p-type IFMRTs, complementary metal-oxide-semiconductor (CMOS) circuits can be built up. Notably, the inherent reconfigurability of IFMRTs endows these circuits with reconfigurable polymorphic functionalities, making them particularly suitable for deployment in hardware security applications. A binary key bit is set to control the reconfiguration of the IFMRTs between n-type and p-type characteristics in the circuit, thereby enabling a singular circuit to execute disparate functions. A quintessential manifestation of this capability is the reconfiguration between NAND and NOR functions in one polymorphic logic gate, which are also the basic logics for any logical operation in a digital circuit[12]. The design paradigm of incorporating such reconfigurable polymorphic logic gates as the standard cell bears resemblance to strategies such as integrated circuit (IC) camouflaging and logic locking, which are pivotal in the field of hardware security[12]. Even if a hypothetical opponent were able to map out the circuit layout by reverse engineering techniques, the transistor polarity and the circuit function remain hidden without the correct key, thus safeguarding the hardware security. In practical implementations, key is a multi-bit binary value. The common key length to meet security standards is 128 bits[42], which offers $2^{128}$ (approximately $3.4 \times 10^{38}$) possible combinations. When sufficient key length is implemented, even a hypothetical opponent cannot computationally determine the specific key value, thereby preventing exposure of the specific circuit function.

To realize reconfigurable polymorphic logic gates, we first implemented an inverter to verify the feasibility of CMOS design with

the schematic delineation presented in Fig. 4a. Since TG has stronger modulation capability, TG acts as PG to control the polarity of IFMRTs so as to obtain more stable n-type characteristics in the circuit, and it is set as key or $\overline{\text{key}}$. The inverter is endowed with two operating modes reconfigured by the key. While key is set to 3 V (logic 1) and $\overline{\text{key}}$ is set to 0 V (logic 0), the upper and lower IFMRTs are p-type and n-type, respectively. While key is 0 V (logic 0) and $\overline{\text{key}}$ is 3 V (logic 1), p-type and n-type IFMRTs are swapped, $V_{dd}$ and GND also do, and the inverter operates in another mode. As depicted by the voltage transfer characteristics (VTC) in Fig. 4b, the inverter can complete the flip between logic 1 and 0 in both modes and achieving gain values of 8.8 and 12.7, respectively. During a bidirectional $V_{IN}$ sweep, it exhibits a clockwise hysteresis behavior (Supplementary Fig. 18). Based on the dual-mode CMOS design, we have realized reconfigurable polymorphic logic gates capable of executing NAND or NOR logic functions, controlled by key, as shown in Fig. 4c. The microscope image of the fabricated logic gates is shown in Fig. 4d and the truth table is shown in Fig. 4e. The curves in Fig. 4f substantiate the functionality of this reconfigurable logic. This NAND/NOR polymorphic logic gate conceals the circuit function, and operates correctly only when the appropriate key is configured, showcasing significant potential for applications in hardware security.

## Homosynaptic characteristics of IFMRT

IFMRTs can also work as synaptic transistors in the burgeoning domain of neuromorphic electronics. Figure 5a displays the information transmission across a biological homosynapse between two neurons. The pre-synaptic pulse travels along the axon to the presynaptic membrane and stimulate the synthesis and release of neurotransmitters, which subsequently bind to the receptors on the postsynaptic membrane. The postsynaptic pulse is generated and travels along the axon. Figure 5b illustrates the analogy between the biological homosynapse and the artificial synaptic device based on a IFMRT. TG is analogous to the presynaptic membrane, the PI-$Al_2O_3$ dielectric imitates the synaptic cleft, and the channel corresponds to the postsynaptic membrane. BG acts as a modulation terminal in the artificial heterosynapse mentioned later, but currently, BG is fixed at 0 V and does not have a modulation effect. The IFMRT exhibits anticlockwise hysteresis characteristic, as illustrated in Fig. 5c, owing to the presence of polar groups in the PI-$Al_2O_3$ dielectric layer, including amino groups with positive charges and carboxyl groups with negative charges[43]. This feature enables the IFMRT to achieve synaptic plasticity. As shown in Fig. 5d, when a negative pulse is applied to TG, the polarization state changes from disorder to order under the action of the electric field. The positively charged groups gather on the TG side and the negatively charged groups gather on the channel side, resulting in an increase in the number of holes in the channel, and leading to an increase of the postsynaptic current (PSC), that is, potentiation behavior. Conversely, applying a positive pulse to TG leads to the decrease in the number of holes in the channel and a decreased PSC, that is, depression behavior. $\Delta I$ is defined as the difference between the current values at the initial state and the moment of pulse withdrawal. Subsequent to pulse withdrawal, the PSC gradually relaxes back to its baseline, corroborating the short-term memory capabilities, mirroring the properties of a biological synapse.

For a single presynaptic pulse, we measured the spike-voltage-dependent plasticity and the spike-duration-dependent plasticity (Supplementary Fig. 19). Taking negative presynaptic stimulation as an example, the pulse with a larger amplitude or width results in the polarization of more dipoles in the dielectric layer, leading to an increased number of holes in the channel. This results in a higher $\Delta I$ and a longer relaxation time. Paired-pulse facilitation shows the potentiation capability of IFMRTs (Supplementary Fig. 20 and Supplementary Note 10). For repeated presynaptic pulses, the spike-frequency-dependent plasticity is presented in Fig. 5e, f

(Supplementary Fig. 21), and the spike-number-dependent plasticity is also presented (Supplementary Fig. 22). For the potentiation behavior, increase of either the frequency or the number of negative pulses causes the accumulation of more holes in the channel, resulting in the increase of $\Delta I$ and the relaxation time. The fundamental synaptic plasticity illustrates the capability of IFMRTs for applications in synaptic transistors and neuromorphic electronics.

## Artificial reconfigurable heterosynapse by IFMRT

Besides the homosynapse between two neurons, heterosynapse involving three neurons also constitutes a critical and integral synaptic connection within an organism. Heterosynapses, characterized by intricate morphologies, are capable of executing sophisticated neural activities, including heterosynaptic plasticity. As shown in Fig. 6a, heterosynapse can be considered to be structurally composed of a homosynapse and a modulatory terminal. The heterosynaptic plasticity means that the synaptic response to the same presynaptic stimulus can be modulated to be stronger or weaker and even reconfigured between inhibitory and excitatory modes by the neuromodulators from the modulatory terminal, and the reconfiguration can be accomplished by the co-release of inhibitory and excitatory neurotransmitters[21,22]. The inhibitory and excitatory modes are defined based on the directionality of the postsynaptic pulse relative to the presynaptic pulse. That is, the inhibitory modes corresponds to opposite directions, while the excitatory mode corresponds to identical directions (Fig. 6b).

Here, by IFMRTs, we demonstrate artificial heterosynapses and achieve the heterosynaptic plasticity, especially the reconfigurable synaptic response, which can be attributed to the multimode modulatory capability and reconfigurability of IFMRTs. As shown in Fig. 6c, BG imitates the modulatory terminal, enabling the modulation and reconfiguration of synaptic responses within the top-gate synaptic transistor by the adjustment of $V_{bg}$. Figure 6d, e exhibit the modulated normalized PSC under different $V_{bg}$ in response to positive and negative pre-synaptic pulses applied to TG, respectively. The variation of $\Delta I$ with modulation voltage $V_{bg}$ clearly indicates the modulation and reconfiguration processes of the synaptic response. In the case of positive pulses in Fig. 6d, for example, when $V_{bg}$ is 0 V, the synaptic response shows the depression behavior with negative $\Delta I$. Increasing $V_{bg}$ in the negative direction (−0.5, −1.0, and −1.5 V) leads to a gradual increase in the absolute value of $\Delta I$, while an increase in the positive direction (0.5 V and 1.0 V) results in a gradual decrease. It indicates that the synaptic responses to the same pluses are either strengthened or weakened by the modulatory signal $V_{bg}$. Continuing to increase $V_{bg}$ beyond 1.5 V transitions the synaptic response from depression behavior to potentiation behavior, with $\Delta I$ becoming positive. As $V_{bg}$ further increases to 3.0 V, $\Delta I$ enlarges, signifying stronger synaptic responses. Consequently, the inhibitory mode is reconfigured to excitatory mode, and the connection strength of the synapse in both of the two modes can be modulated by $V_{bg}$. As shown in Fig. 6e, a similar situation also occurs in the case of negative pre-synaptic pulses. When $V_{bg}$ is 0 V, the synapse is inhibitory mode, exhibiting potentiation behavior in response to negative pre-synaptic pulses. Adjusting $V_{bg}$ between −0.5 V and 1.0 V modulates the synapse between stronger and weaker connections. With a larger positive voltage applied to $V_{bg}$, the synapse is reconfigured to excitatory mode, exhibiting depression behavior. The dependencies of $\Delta I$ with $V_{bg}$ for both cases of positive and negative pre-synaptic pulses are extracted and presented in Fig. 6f-g, respectively. It elucidates that the same pre-synaptic stimulus can induce responses with various directions and strength under the modulation effect. That is, the flexible modulation and reconfiguration of synaptic properties are realized by IFMRTs, with BG serving as a pivotal modulatory terminal in this process.

The mechanism of modulated and reconfigured synaptic characteristics is elucidated through an analysis of the PSC and the charges

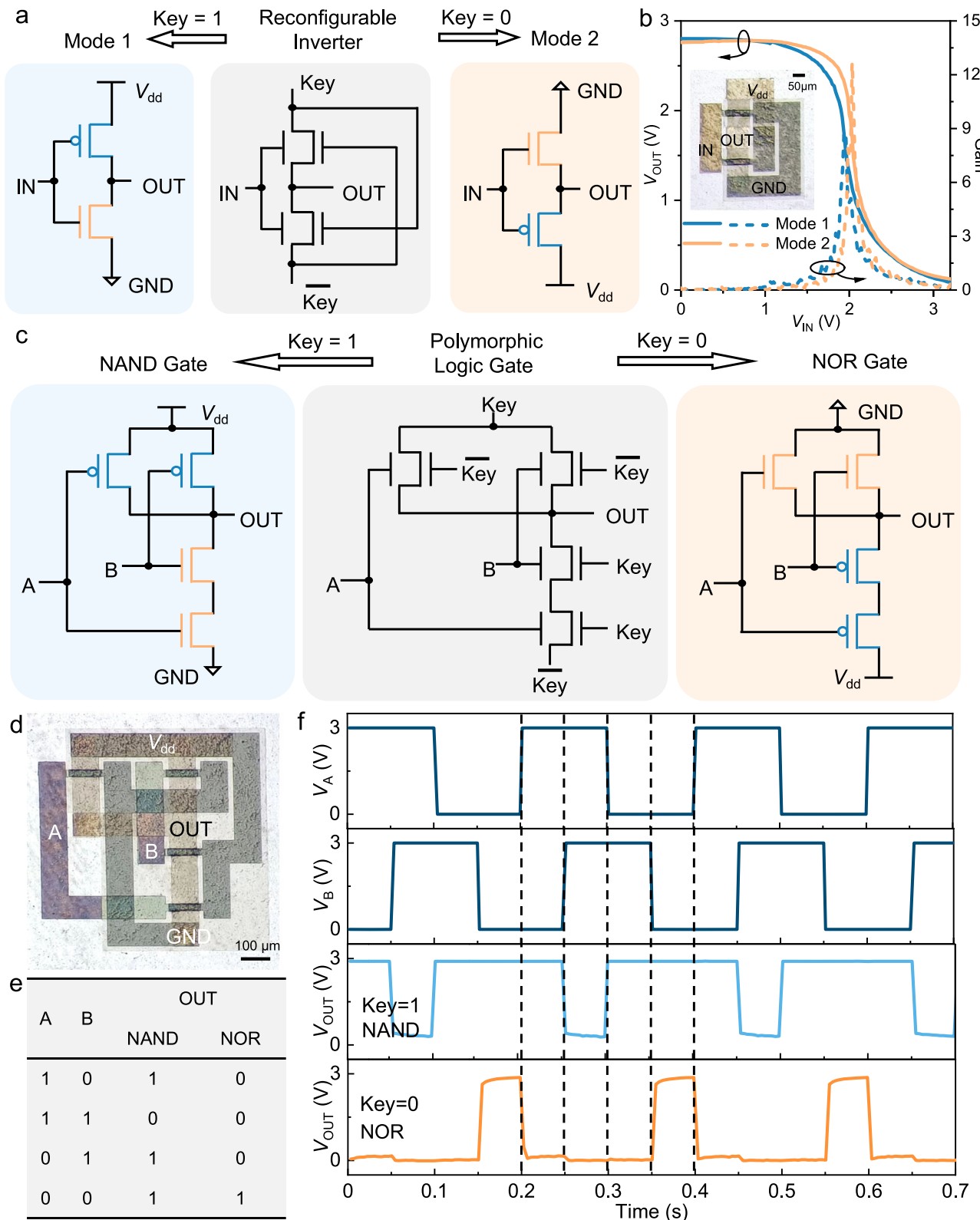

**Fig. 4 | Reconfigurable inverters and polymorphic logic gates. a, b** Schematic explanation (**a**) and VTCs (**b**) of the reconfigurable inverter operating in two modes. Inset, the optical microscope image of the inverter. W and L of the IFMRTs are 100 μm and 20 μm, respectively. **c** Schematic explanation of the reconfigurable polymorphic logic gates. The two upper and lower transistors are connected in parallel and in series, respectively, and all the TG terminals are set as key or $\overline{\text{key}}$. The two operating modes are NAND gate when key = 1 and NOR gate when key = 0. **d, e** Optical microscope image (**d**) and truth table (**e**) of the reconfigurable polymorphic logic gates. **f** The output waveforms of the two operating modes, NAND gate and NOR gate.

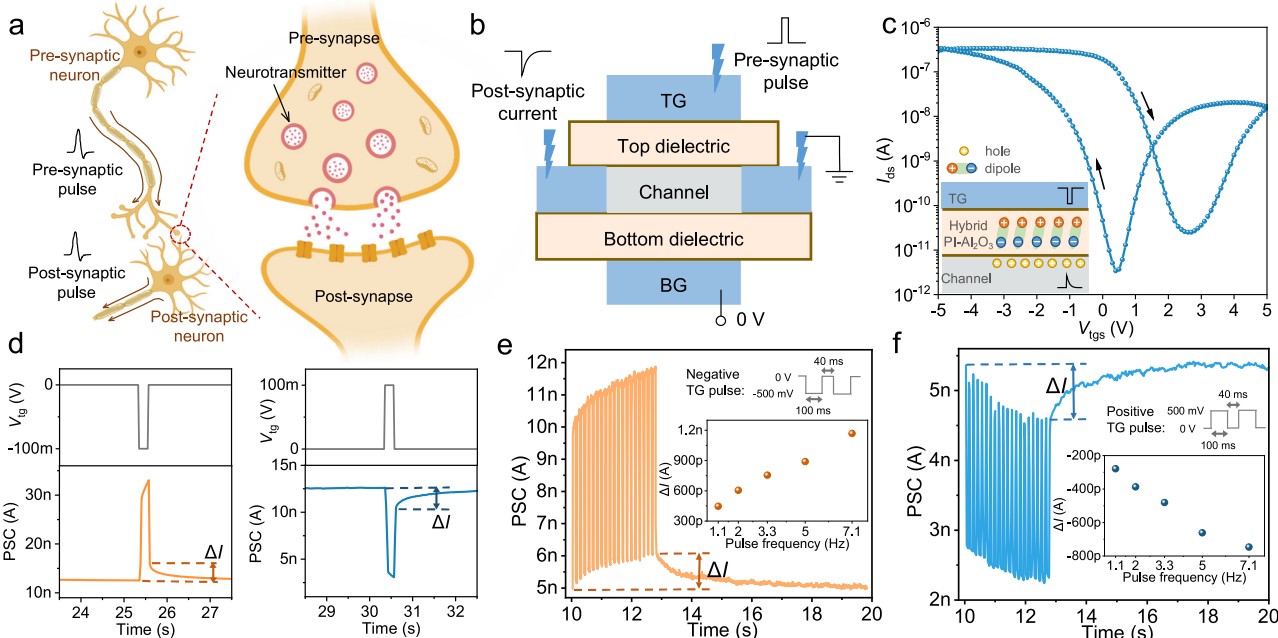

**Fig. 5 | Homosynaptic characteristics of IFMRT. a** The biological structure of a homosynapse between two neurons. **b** Schematical illustration of the artificial homosynaptic device based on IFMRT. **c** Hysteresis curve of a IFMRT when $V_{tg}$ is scanned and $V_{bg}$ is set to 0 V. The inset shows the schematic diagram of the synaptic mechanism when a negative pulse is applied to TG. ($V_{ds} = -0.1$ V, W = 50 μm, L = 50 μm) **d** Typical potentiation and depression behaviors triggered by a negative and positive presynaptic pulse, respectively. ($V_{ds} = -0.1$ V, W = 50 μm, L = 50 μm) **e**, **f** Potentiation (**e**) and depression (**f**) behaviors triggered by multiple negative and positive presynaptic pulses, respectively. The inset shows the relationship of $\Delta I$ and pulse frequency ($V_{ds} = -0.1$ V, W = 50 μm, L = 50 μm). PSC is the abbreviation of postsynaptic current. (The nerve cell and synapse in (**a**) are created with MedPeer (medpeer.cn)).

within the dielectric layer. Taking the case of positive presynaptic pulses as an example, the PSC under different $V_{bg}$ with positive pulses applied on TG are shown in Fig. 6h–i. The schematic diagrams of the internal charges are shown in Fig. 6j. When the synapse works in inhibitory mode, as $V_{bg}$ set to −0.5, 0, or 0.5 V, majority carriers in the channel are holes and the positive TG pulses lead to the depression behavior. In this process, a larger $V_{bg}$ is correlated with a reduced concentration of holes in the channel, as evidenced by a decreased base current of PSC. Therefore, as $V_{bg}$ becomes larger, the same TG pulse stimulus leads to a smaller reduction in holes, causing a smaller absolute value of $\Delta I$ and a weaker synaptic response in inhibitory mode. When the synapse in excitatory mode, as $V_{bg}$ set to 2.0, 2.5, or 3.0 V, majority carriers in the channel are reconfigured to electrons. The positive TG pulses also lead to the ordered polarization state, inducing electron accumulation and the corresponding potentiation behavior, which illustrates the reconfigurable synaptic responses. Moreover, the increased $V_{bg}$ modulates the number of electrons in the channel, causing an increase in the base current of PSC. Consequently, the same presynaptic pulse leads to a larger increase in electrons, causing a larger $\Delta I$ and a stronger synaptic response in excitatory mode. A similar analysis can also be applied to the case of negative presynaptic pulses at different $V_{bg}$ (Supplementary Fig. 23).

**Dendrite integration by IFMRT for robotic decision and motion**
In biological neural networks, neurons exhibit multiple dendrites, each capable of receiving signals from other neurons, integrating the information, and subsequently transmitting the signals to other neurons, as shown in Fig. 7a. The dendrite integration enhances the information reception and processing of neural networks at the individual neuron level[27,28]. Here, we have simulated dendrite integration based on IFMRT and realized artificial dendrite. In IFMRT, as shown in Fig. 7b, TG and BG both serve as the pre-synapses while the channel works as the dendritic postsynaptic membrane. The previous section

illustrates heterosynaptic plasticity, in which the presynaptic neuron is the dominant input and the modulatory neuron works as a modulator. This section illustrates dendrite integration, in which both presynaptic membranes work as inputs equally. The different coupling capabilities of BG and TG provide greater freedom and plays a crucial role in achieving dendrite integration, which holds considerable potential for applications in intelligent robotic decision and robotic arm control.

Firstly, for the robotic decision, intelligent robots commonly encounter multiple external signals and need to make correct judgments according to these signals. Taking two binary signals as an example, the possible input states are 00, 01, 10, and 11. Here, we have achieved the complete distinction of the four states with only one IFMRT to illustrate its application potential in automatic obstacle avoidance for robots. The diagram of robotic obstacle avoidance is exhibited in Fig. 7c. 1 and 0 refer to the presence and absence of an obstacle on the robot's any one side, respectively. The two commercially available collision sensors are connected to TG and BG, respectively, and they output a −0.5 V voltage when pressed. The photo and schematic diagram of the testing are shown in Fig. 7d, e. Corresponding to Fig. 7c, we controlled the waveforms of TG and BG by pressing the button of the collision sensors, and the performance of the IFMRT-based artificial dendrite is shown in Fig. 7f. The four states are one-to-one corresponded in the two figures. TG and BG represent the robot's left and right sensory inputs, respectively. A negative pulse with the amplitude of −500 mV signifies the detection of an obstacle, and no pulse indicates a clear path. The postsynaptic current reflects the robotic decisions regarding obstacle avoidance in the four states. Absence of pulse input to both BG and TG simulates an unobstructed environment, with the postsynaptic current maintaining at a baseline level, suggesting the robot's continuing forward motion. When a pulse is applied to either BG or TG, the postsynaptic current exhibits a positive pulse with the peak value of about 22 nA or 62 nA. The higher coupling capability of TG leads to the higher postsynaptic pulse,

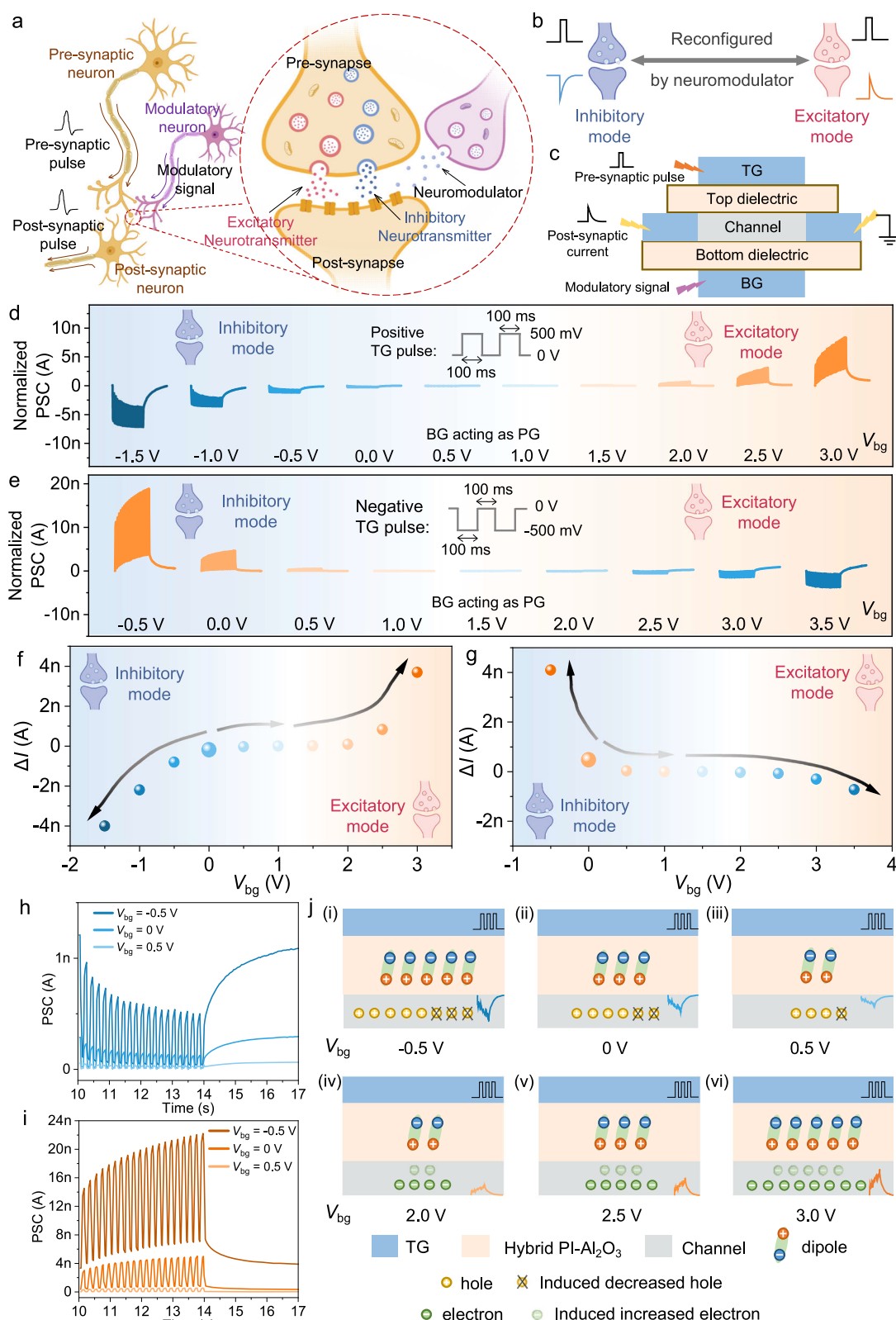

**Fig. 6 | Heterosynaptic plasticity and reconfigurable synaptic characteristics of the artificial IFMRT heterosynapse. a** Biological structure of a heterosynapse among three neurons. **b** Schematic diagram of the reconfigurability between inhibitory and excitatory modes. **c** Schematic illustration of the analogy of pre-synapse, post-synapse, and modulatory neuron in artificial heterosynapse based on IFMRT. **d, e** The modulated and reconfigured normalized PSCs under different $V_{bg}$ when positive (**d**) and negative (**e**) pre-synaptic pulses are applied on TG. ($V_{ds} = -0.1$ V, W = 50 μm, L = 50 μm) **f, g** Relationship between $\Delta I$ and $V_{bg}$ when positive (**f**) and negative (**g**) pre-synaptic pulses are applied on TG. **h–i** PSCs in inhibitory mode (**h**) and excitatory mode (**i**) at different $V_{bg}$ when positive pre-synaptic pulses are applied on TG. ($V_{ds} = -0.1$ V, W = 50 μm, L = 50 μm) **j** The schematic diagrams of the internal charges at different $V_{bg}$. PSC is the abbreviation of postsynaptic current. The synapse icons in (**d–g**) are defined in (**b**). (The nerve cell and synapse in (**a**), as well as the synapse icons in (**b** and **d–g**), are created with MedPeer (medpeer.cn)).

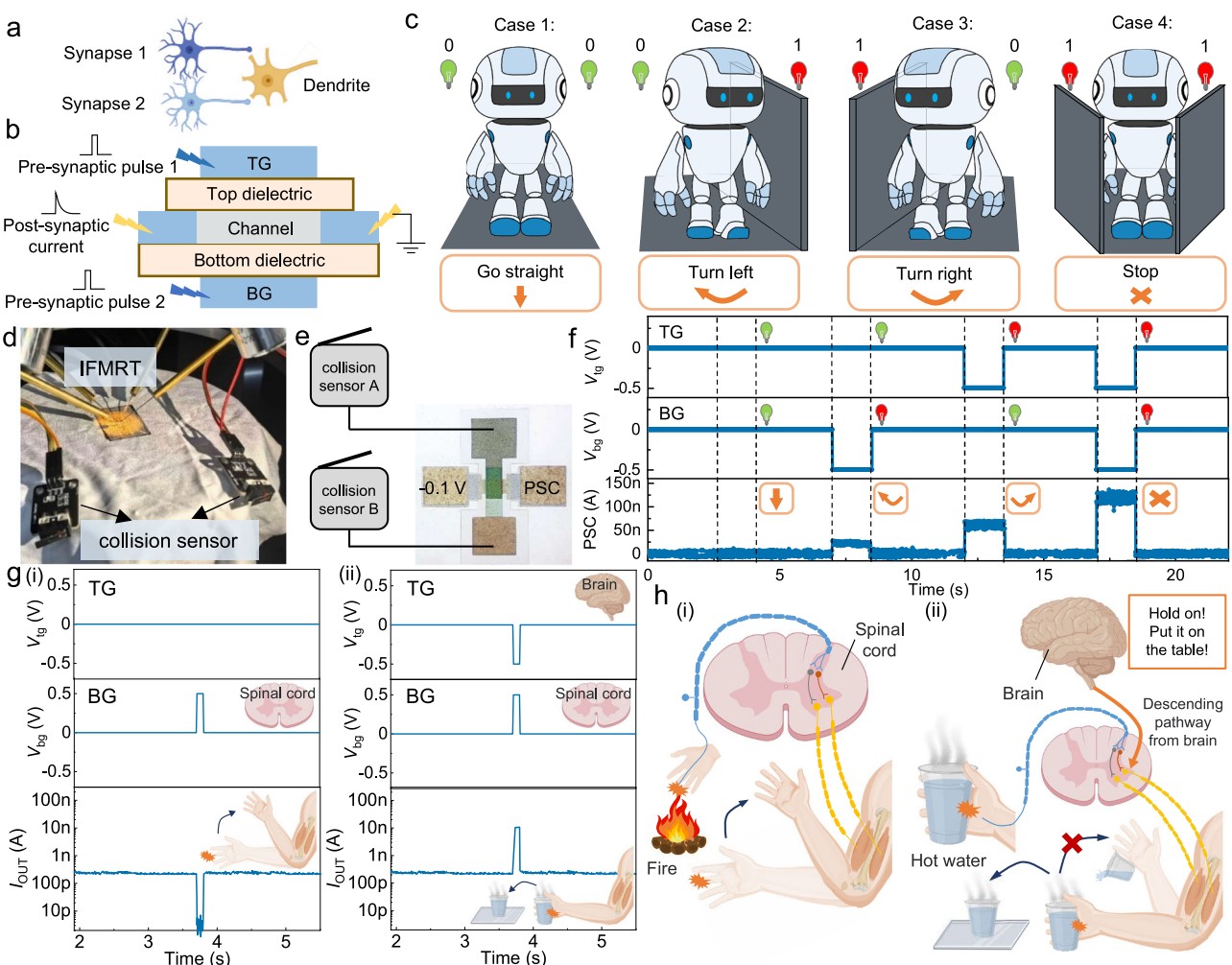

**Fig. 7 | Dendrite integration by IFMRT for robotic decision and motion.**
**a** Biological structure of the dendrite in neural network. **b** Schematic illustration of the dendrite integration based on the IFMRT. **c** The diagram of robotic obstacle avoidance in the four situations, including Go straight, Turn left, Turn right and Stop. **d**, **e** the practical test photo (**d**) and the schematic diagram (**e**) of the connected collision sensors and an IFMRT. **f** The PSC when four different combinations of pulses are applied to BG and TG, simulating the corresponding four situations of robotic obstacle avoidance. ($V_{ds}$ = −0.1 V, W = 50 μm, L = 50 μm) **g** PSCs when (i) only a positive pulse is applied to BG and (ii) a pair of opposite pulses is applied to BG and TG, simulating the neural activities controlled only by the spinal cord and by both of the spinal cord and brain, respectively. ($V_{ds}$ = −0.1 V, W = 50 μm, L = 50 μm) **h** (i) Diagram of the hand withdrawal reflex in response to a thermal stimulus, controlled by the lower-level neural center, spinal cord; (ii) The diagram of the hand holding the cup and placing it on the table quickly instead of throwing it away, controlled by the higher-level neural center, brain. PSC is the abbreviation of postsynaptic current. The light bulb icons in (**f**) are defined in (**c**). (The spinal cord, brain, arm, cup, glass, and nerve cell in (**g**, **h**) are created with MedPeer (medpeer.cn)).

leading the robot turning right when it encounters an obstacle on the left. The smaller peak represents the decision of turning left when the robot detects an obstacle on the right. Simultaneously applying pulses to both BG and TG results in a peak value of about 120 nA. The collaborative control exerted by both TG and BG over the channel produces the highest current peak, indicating that the robot will cease movement when obstacles are present on both the left and right sides. Therefore, the artificial dendrite based on a single IFMRT proves its capability of completely distinguishing the four logical states. It is not only suitable for the intelligent robot obstacle avoidance scenarios, but also promising for applications in any scenario involving two binary inputs to integrate information and make judgments.

The dendrite integration based on IFMRT also has high application potential in the robotic arm control of humanoid robots. Using the concept of bionics, the robotic arm can be intelligentized by imitating the human motor system, which is governed by the central nervous system, encompassing both lower-level neural centers, such as the spinal cord and higher-level neural centers, such as the brain. Here, we have mimicked the cooperative control mechanism in the human motor system by IFMRT-based dendrite integration. The stronger channel coupling effect of TG emulates the higher-level control ability of the brain, while the weaker channel coupling effect of BG emulates the lower-level control ability of the spinal cord. As shown in Fig. 7g (i), applying a positive pulse exclusively to BG results in a depression postsynaptic response, representing the autonomous withdrawal reflex controlled independently by the spinal cord in response to a thermal stimulus, as shown in Fig. 7h (i). Brain, commanding a higher hierarchical position in motion control, can override the spinal cord's reflexive actions, as illustrated in the scenario of Fig. 7 (ii). Here, an inadvertent grasp of a hot cup prompts the higher-level brain to counteract the spinal withdrawal, controlling the arm to temporarily hold the cup and place it on the table as quickly as possible, instead of

throwing it away. For the IFMRT-based artificial dendrite, as shown in Fig. 7g (ii), the simultaneous application of positive and negative pulses to BG and TG, respectively, represents the concurrent influence of the spinal cord and the brain as observed in the scenario of Fig. 7h (ii). The resulting postsynaptic potentiation response, due to the stronger coupling ability of TG, simulates the higher-level control authority of the brain, directing the arm to perform the action of holding the cup and placing it on the table. This biomimetic emulation of coordinated motion control by the artificial dendrite, integrating the functions of lower- and higher-level neural centers, equips robots with the capability to adopt different coping strategies for diverse situations, thereby enhancing the subtlety and adaptability of robotic motion control. The control of low-level neural centers and related foundational conditional reflexes, simulated by BG, serves to protect robots from external hazards such as heat or sharp objects. The high-level neural centers, simulated by TG, can break away from the singleness of conditional reflexes and provide the control capability in higher hierarchy to cope with complex situations.

To sum up, the dendrite integration by IFMRT takes advantages of the differentiation of two gate electrodes, which holds significant potential for applications in robotic decision, robotic arm control, and intelligent prosthesis. This proof-of-concept establishes a crucial groundwork for future implementations of more complex and adaptive device behaviors with higher-level flexible integration and multimode reconfigurable capabilities. The IFMRT acts as a fundamental enabler, which holds the potential to take the system concepts demonstrated on rigid, segmented hardware[44,45] to the realm of efficient, integrated, and truly biomimetic robotic nervous systems.

## Discussion

We have reported IFMRTs and the demonstration in reconfigurable circuits, artificial heterosynapse, and dendrite integration. The dual-gate structure and the ambipolarity of the carbon-nanotube channel doped by $Al_2O_3$ bring the multimode and multithreshold-voltage reconfigurability among p-type, n-type, and ambipolar modes, which can be controlled by either of the two gates. One gate serves as the modulation terminal, while the other one switch the transistor on and off. The reconfigurability can be theoretically explained by the modulated surface energy band and the initial channel conductance. Based on the reconfigurable transistors, we demonstrate the polymorphic inverters and logic circuits, which provides a potential for hardware security. In terms of neuromorphic devices, artificial heterosynapses are realized and the heterosynaptic plasticity, especially the reconfigurable synaptic responses, are obtained. Besides, the dendrite integration is also achieved. We demonstrate the solutions for the automatic obstacle avoidance and the coordinated movement control by low-level and high-level neural centers. These demonstrations indicate the application potentials of IFMRTs in intelligent robotic decision and arm control. Besides, IFMRT array owns excellent flexibility, which can withstand 5000 cycles of bending at a radius of 4 mm. Besides, IFMRT can withstand 5000 cycles of bending at a radius of 4 mm, shows excellent flexibility which are seldom reported before. The proposed IFMRT opens opportunities for flexible wearable electronics and soft artificial intelligent robotic systems.

## Methods

### Device and circuit fabrication
IFMRTs and circuits were fabricated on PEN substrates. Before the fabrication, the substrate was baked at 180 °C for 15 min and oxygen plasma treated at 100 W for 5 min. The BG electrodes were fabricated by spin-coating 0.5 wt.% M-CNT suspension (Chinese Academy of Sciences Chengdu Organic Chemistry Co. Ltd.), photolithography patterning and oxygen plasma etching. The plasma etching was performed using a gas mixture of nitrogen (3 L/min) and oxygen (5 cm³/min) at a power of 100 W. The duration was 35 minutes for M-CNTs. The first dielectric layer, hybrid PI-$Al_2O_3$, was deposited by ALD and MLD technique. The thickness of the dielectric in our work is 38 nm consisting of 2-nm $Al_2O_3$ (bottom buffer layer), 30-nm PI-$Al_2O_3$ hybrid film and 5-nm $Al_2O_3$ (top buffer layer). The S/D electrodes were prepared on the first dielectric layer by spin-coating M-CNT suspension, followed by photolithography patterning and 35-minute oxygen plasma etching. After that, S-CNT suspension was spin-coated and patterned by photolithography and 20-min oxygen plasma etching to form the channel. The SCNT suspension consisted of 0.01 mg mL$^{-1}$ SCNT, which was prepared by ultrasonicating 0.5 mg SCNT (NanoIntegris Inc., 99.9%) and 0.5 mg poly[(m-phenylenevinylene)-co-(2,5-dioctoxy-p-phenylenevinylene)] (PmPV) in 50 mL 1,2-dichloroethane for 6 h. Then, the second 38-nm hybrid PI-$Al_2O_3$ dielectric was deposited on the channel layer by the same process as the first one. The TG electrodes were prepared on the dielectric layer by spin-coating and patterning M-CNT layer. Consistent with BG patterning, TG electrodes were patterned by photolithography and oxygen plasma etching. BG and S/D pads were opened by 85% phosphoric acid etching with patterned photoresist. The fabrication process of circuits is similar to the abovementioned process of IFMRTs, except that the contact holes were opened before the preparation of BG and S/D layers for connections between different layers.

### Device and circuit characterization
The electrical characterizations of IFMRTs and circuits were performed by an Agilent B1500A semiconductor analyzer at room temperature. The input signal of circuit dynamic characterizations and synaptic behavior was generated by a pulse generator (16440 A, Agilent). Capacitance was measured by an LCR meter (E4980A, Keysight). The XPS analysis of $Al_2O_3$/CNT thin-film samples coated on silicon wafers is performed by using Escalab 250Xi XPS system (Thermo Scientific) with an Al Kα source. Optical transmittance is measured by UV-2600 spectrophotometer from SHIMADZU CORPORATION. Device bending test was conducted on a digitally controlled mechanical stage.

### Hybrid PI-$Al_2O_3$ fabrication
The hybrid PI-$Al_2O_3$ films were deposited in a home-built tubular ALD/MLD reactor. The polyimide were deposited by MLD from 1,2-ethylenediamine (EDA) and pyromellitic dianhydride (PMDA) as the precursors, and the hybrid PI-$Al_2O_3$ films were deposited by alternately MLD of the polyimide and ALD of $Al_2O_3$ from trimethylaluminum (TMA) and water vapor. During deposition, EDA, TMA, and water were kept at room temperature (25 °C), while PMDA was heated to 163 °C. Purified $N_2$ gas was used as the carrier gas to assist the precursor delivery, and it was also used as the purge gas. The chamber pressure during purging was approximately 1 Torr. The deposition temperature was at 165 °C, which was higher than the PMDA vaporization temperature to avoid its condensation. The MLD of the polyimide was conducted in closevalve mode. 2 pulses of EDA vapor and 10 pulses of PMDA vapor were dosed in each MLD cycle, which corresponded to the exposures of 0.033 and 0.063 Torr s for EDA and PMDA, respectively. Long purge time of 100 s was employed to ensure complete removal of the byproduct and excess precursors. The ALD of $Al_2O_3$ was processed in flow-through mode, where the exposures of TMA and water vapor in each ALD cycle were 0.014 and 0.064 Torr s, respectively. The hybrid MLD/ALD process was performed in a supercycle approach, where each supercycle consisted of 3 MLD cycles of the polyimide and 3 ALD cycles of $Al_2O_3$.

## Data availability
The data generated in this study are provided in the Source Data file. Source data are provided with this paper.

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

## Acknowledgements

This work was financially supported by CUHK-Shenzhen University Development Fund (No. UDF01003868, M.Z.), National Key Research and Development Plan (No. 2022YFB3603600, M.Z.), and National Natural Science Foundation of China (No. 62074008 M.Z. and No. 22175005 X.W.).

## Author contributions

M.Z. conceived and supervised the project. W.W. fabricated devices and circuits. J.Z., J.W. fabricated the dielectric. W.W., R.Q., and Q.H. conducted device design and circuit design. W.W., R.Q., Q.H., X.W., and M.Z. analyzed the data and mechanism. W.W. and T.Z. performed electrical and mechanical characterizations. D.L. and Z.W. characterized XPS. W.W., J.L., C.W., and S.Z. performed demonstrations. W.W., R.Q., and M.Z. wrote the manuscript, which was approved by all co-authors.

## Competing interests

The authors declare no competing interests.
