## [Transparent Peer Review file · Nature Communications]

Intrinsically Flexible Multimode Reconfigurable Transistors for Polymorphic Circuits and Neuromorphic Devices

Corresponding Author: Professor Min Zhang

Version 0:

Reviewer comments:

Reviewer #1

(Remarks to the Author)

Major revision is recommended.

The authors proposed an intrinsically flexible multimode reconfigurable transistors (IFMRTs) with a dual-gate structure, capable of operating in p-type, n-type, and ambipolar modes. These transistors can be modulated by either gate, allowing for adjustable threshold voltages. Demonstrations include inverters and polymorphic logic circuits with selectable NAND or NOR functions, offering solutions for hardware security. Additionally, IFMRTs enable artificial heterosynapse and dendrite integration, facilitating reconfigurable synaptic responses. These devices maintain their reconfigurability after 5000 bending cycles at a 4 mm radius, showing potential applications in flexible wearable electronics and intelligent robotic systems. The proposed dual gate based reconfigurable device system is interesting. However, I have major concerns regarding the applicability of the proposed heterosynapse and dendrite integration:

1. If PI-Al₂O₃ hybrid dielectric have a hysteresis effect on CNT semiconducting channels as shown in Figure 5c, how can they be used as logic circuits? Typically, logic circuits for computing require negligible hysteresis. Hysteresis behaviors should be shown in Figure 2 and 3.
2. The retention behavior of the device does not seem good. PPF and nonlinearity data or other additional data that can provide synaptic performance of the device is required. The device may lack synaptic performance due to the intrinsic limitation of the material (typically ionic dielectric interfaced with ion penetrable channels possess good retention behaviors).
3. In Figure 6e and g, the notation of 'inhibitory mode' and 'excitatory mode' is confusing. With $PG = -0.5 V$, the PSC is increasing, shouldn't this be the 'excitatory mode'? Also, the modulation by PG is asymmetric for positive and negative TG pulses. This may limit the application of the device, as synaptic strengths relative to positive and negative presynaptic signals may differ substantially at a given PG voltage.
4. In Figure 7, robotic motion control scenario is rather speculative. More complicated circuits and systems may be required to actuate the proposed signals into real robots hindering the applicability of the device. It is advised that a more realistic demonstration be provided.
5. In Figure 7e, the BG signal of (i) and (ii) are opposite signs. If a similar or identical thermal stimuli is applied shouldn't the BG (equivalent to spinal cord) signal be in the same sign?

Reviewer #2

(Remarks to the Author)

I attached my review.

Reviewer #3

(Remarks to the Author)

The paper describes the implementation of intrinsically flexible multimode reconfigurable transistors (IFMRTs) with dual-gate structure. Thanks to the modulation of the transistor modes, n- and p-types, as well as ambipolar, the IFMRTs are implemented for inverters and logic circuits realization, and as neuromorphic devices, achieving synaptic response. In addition, engineered solutions to tackle robotic motion and arm movement are reported. We demonstrate the solutions for the automatic obstacle avoidance and the coordinated movement control by low-level and high-level neural centers.

The following points should be clarified to edit the paper and improve the overall quality:

1. What is the state-of-the-art related to “reconfigurable transistors” and “flexible reconfigurable transistors”? This point plays a crucial role to understand how this manuscript can impact the scientific community;
2. The device fabrication (page 3, paragraph “Methods” and Supplementary figure 1) is not described in details. In particular:
 - No information is provided for the “M-CNT suspension”;
 - No information is provided for the oxygen plasma etching (power, time, etc.);
 - No information is provided for the “S-CNT suspension”;
 - Which kind of patterning is employed for the Tog Gate electrode?
 - What is the concentration for the phosphoric acid for the etching of BG and S/D pads?
3. Can the authors explain the definition of “intrinsically flexible layer design for the dielectric”? This concept is quite unclear for the lack of proper explanation in the manuscript and for the lack of reference 41 in the “reference” paragraph;
4. The authors claim that the IFMRT array shows “good transmittance”. Can they provide precise transmittance value in the visible spectrum (layer by layer and for the whole device stuck)? In addition, they should clarify how the Supplementary Figure 2 is acquired: what “without substrate” means? Was the sample with the array used for the transmittance measurement? Or was there a new sample with the unstructured layers deposited one on top of each other used to simulate the transmittance through the whole device stuck? This will give a better feeling to the reader on the transmittance of the devices;
5. The authors claim “The ambipolarity of the S-CNT channel arises from the n-type doping of Al₂O₃ in the upper and lower dielectric layers.”. Can they provide scientific explanation of this doping? Have they performed any material analysis of the dielectric layer to support this statement?
6. What is the leakage gate current in Figure 2b and 2c? These trends should be reported also in these plots for the sake of comprehensiveness (besides Supplementary Information);
7. Can the authors provide scientific support to “During the oxygen plasma etching the S/D electrodes, the upper surface of the bottom dielectric layer suffers minor damage, leading to the weaker coupling capability of BG.” ? Was there any damage visible on the dielectric surface? More details should be provided;
8. In the Reference paragraph, there are 40 references, whereas, the authors use 42 references in the manuscript;

Version 1:

Reviewer comments:

Reviewer #1

(Remarks to the Author)

While the authors have provided a response, several important concerns remain regarding the practical implementation of the proposed device and system. Firstly, the device is intended to function both as a conventional logic circuit and as a synaptic transistor. However, synaptic transistors typically rely on the slow relaxation dynamics of the PI-Al₂O₃ dielectric, which inherently limits switching speeds. This trade-off raises concerns about the suitability of the device for conventional high-speed binary logic operations, potentially restricting its applicability in real electronic systems.

Furthermore, in Figure 7, the demonstration of robotic control appears limited in scope, as the robotic movements are predetermined. Real-world robotic applications typically require more complex and adaptive device behaviors. For reference, more advanced demonstrations can be found in works such as Nature Communications 14, 5 (2023) and Science Advances 8, eabo3326 (2022), where device functionalities are more closely aligned with the demands of dynamic and responsive robotic systems.

In the context of this work, the advantages of synaptic plasticity appear to be underutilized or unnecessary for the demonstrated application. Therefore, the practical value of employing a multifunctional device in this manner may be limited.

Reviewer #2

(Remarks to the Author)

I conducted a thorough examination of the revised manuscript and SI. I am satisfied that the response letter adequately addresses all of the concerns I raised. and the manuscript and SI are well revised. I was grateful for the author's dedication to improving the manuscript's quality.

Reviewer #4

(Remarks to the Author)

Manuscript title: Intrinsically flexible multimode reconfigurable transistors for polymorphic circuits, artificial heterosynapse,

This manuscript reports the development of an intrinsically flexible multimode reconfigurable transistor (IFMRT), integrating dual-gate control, ambipolar operation, and dendrite-inspired neuromorphic functions. The authors demonstrate polymorphic logic gates, reconfigurable synaptic plasticity (including artificial heterosynapse behavior), and dendritic integration for robot decision-making and motor control, all using a single device architecture. The concept is novel and well-motivated, and the experimental demonstrations are convincing and multidisciplinary in scope, spanning circuit applications, synaptic behavior emulation, and robot control.

The manuscript is generally well written and logically structured. I believe this work has strong potential for publication. However, a few important clarifications and additions are required before I can recommend acceptance.

Major Comments

1. Transistor Operation Regime and Output Characteristics

All transistor characterizations appear to be performed in the linear regime (e.g., $V_{ds} = -0.5$ V), and no data are shown for the saturation regime (e.g., $V_{ds} = -3$ V), which is essential for digital logic and analog gain evaluation. It would strengthen the manuscript to include output characteristics in the saturation region and to discuss saturation current, output resistance, or intrinsic gain if available.

2. Mobility Asymmetry and Circuit-Level Implications

Supplementary Figure 11 shows that the extracted field-effect mobility differs significantly between p-type (7–14 cm^2/Vs) and n-type (2.5–7.5 cm^2/Vs) modes when TG acts as the programming gate, and this difference becomes even larger when BG is used as the programming gate.

(1) Could this mobility asymmetry be responsible for the observed shift of the inverter switching point toward the positive side of $V_{dd}/2$ in Fig. 4b?

(2) In conventional complementary logic circuits, mobility mismatch is typically compensated by adjusting the W/L ratio of p- and n-type transistors. However, the main advantage of IFMRTs lies in their reconfigurability — the ability to define p- or n-type behavior after fabrication. If W/L must be predetermined to account for this asymmetry, does this not undermine that advantage?

(3) In this context, do the authors consider the observed mobility imbalance and resulting shift in logic thresholds to be within acceptable margins for future high-density integration?

Additionally, was the inverter characteristic in Fig. 4b obtained from a single-sweep measurement or during a bidirectional V_{in} sweep? Is there any observable hysteresis?

3. Mobility Extraction Methodology

While mobility values are provided in Supplementary Figure 11, the extraction method is not described in the manuscript or supplementary information. Please include a brief explanation of the calculation procedure (e.g., equations used, assumptions about gate capacitance or threshold extraction), even if mobility is extracted from the linear regime.

4. Device Geometry and Scaling Discussion

The channel length variation ($L = 20, 50, 100$ μm) is mentioned in the Supplementary Information, but the channel width (W) and W/L ratios used in the main device demonstrations are not clearly stated.

Please specify the W/L ratios for devices used in logic circuits and neuromorphic applications.

Recommendation: Minor Revision

The manuscript introduces a promising class of flexible, reconfigurable transistors and demonstrates their use in logic and neuromorphic applications, including robot control. However, critical points related to mobility asymmetry, scalability, and logic-level design must be addressed to fully support the claims and enable broader applicability. I encourage the authors to consider the above comments carefully to strengthen the manuscript.

Version 2:

Reviewer comments:

Reviewer #1

(Remarks to the Author)

The authors fully addressed the comments raised by all the reviewers.

So, this paper can be published in Nature Communications without further revisions.

Reviewer #4

(Remarks to the Author)

We appreciate the authors' dedicated efforts and the comprehensive revisions made to the manuscript and the Supplementary Information. All key concerns raised during the review process have been addressed in a scientifically rigorous and transparent manner.

Specifically:

1. Transistor Characteristics: The addition of output characteristics, saturation current, output resistance, and intrinsic gain in Supplementary Note 5 and Figure 5 is highly valuable and strengthens the evaluation of both the digital and analog potential of the IFMRTs.

2. Mobility Asymmetry and Scalability: The transparent discussion of the mobility mismatch between p-type and n-type operation, the resulting shift in the logic threshold, and the current noise margin ($\approx V_{DD}/6$) is commendable. The proposal of contact engineering (Supplementary Note 8) as the critical path for future optimization is a crucial and well-articulated strategy for high-density integration.

3. Technical Details: The inclusion of the mobility extraction methodology (Supplementary Note 7), as well as the specific W and L values in the manuscript, significantly improves the reproducibility and clarity of the work.

4. Robotic Application: The clarification that the robotic demonstration serves as a proof-of-concept for the device's foundational reconfigurability and intrinsic flexibility is satisfactory, aligning the scope of the work with its core device innovation.

The manuscript presents a highly innovative and important advancement in the field of intrinsically flexible, multifunctional electronics. The demonstrated integration of reconfigurable logic and neuromorphic functions in a single flexible platform is novel and holds strong potential for future soft robotics and wearable technologies.

The manuscript is now recommended for acceptance.

Point-by-point responses to reviewer comments

Reviewer #1:

Comments:

Major revision is recommended.

The authors proposed an intrinsically flexible multimode reconfigurable transistors (IFMRTs) with a dual-gate structure, capable of operating in p-type, n-type, and ambipolar modes. These transistors can be modulated by either gate, allowing for adjustable threshold voltages. Demonstrations include inverters and polymorphic logic circuits with selectable NAND or NOR functions, offering solutions for hardware security. Additionally, IFMRTs enable artificial heterosynapse and dendrite integration, facilitating reconfigurable synaptic responses. These devices maintain their reconfigurability after 5000 bending cycles at a 4 mm radius, showing potential applications in flexible wearable electronics and intelligent robotic systems. The proposed dual gate based reconfigurable device system is interesting. However, I have major concerns regarding the applicability of the proposed heterosynapse and dendrite integration:

Authors' response:

We thank the reviewer for the positive summary of our work.

1. If PI-A1₂O₃ hybrid dielectric have a hysteresis effect on CNT semiconducting channels as shown in Figure 5c, how can they be used as logic circuits? Typically, logic circuits for computing require negligible hysteresis. Hysteresis behaviors should be shown in Figure 2 and 3.

Authors' response:

Thanks you for the valuable comments. Yes, the traditional transistors used for logic circuits expect negligible hysteresis ideally. However, previous research has shown that circuits such as NAND, NOR gate, 101-stage ring oscillator can operate well even with a hysteresis of around 2 V^{R1}. That is, it is feasible to implement circuits using devices that exhibit certain hysteresis. Besides, we can modulate the hysteresis by adjusting the dielectric process, which was reported by our previous work in Fig. 2 of Ref. R2. **We have updated the corresponding description in the revised manuscript to make it clearer.**

We have supplemented the hysteresis curves in Fig. 2m based on your advice.

References

- R1 Wang, B. et al. Continuous fabrication of meter - scale single - wall carbon nanotube films and their use in flexible and transparent integrated circuits. *Adv. Mater.* **30**, 1802057 (2018).
- R2 Zhu, J. et al. Ultra-Flexible High-Linearity Silicon Nanomembrane Synaptic Transistor Array. *Adv. Mater.* 2413404 (2025).

Changes: *Yes*

Location of Change in manuscript:

Section: **Device structure and characteristics of the IFMRT**

Page: *Page 6, Fig. 2m; Page 7, Paragraph 1 Line 19 (green highlighted)*

Section: **References**

Page: *Page 20, Ref 42 (green highlighted)*

2. The retention behavior of the device does not seem good. PPF and nonlinearity data or other additional data that can provide synaptic performance of the device is required. The device may

lack synaptic performance due to the intrinsic limitation of the material (typically ionic dielectric interfaced with ion penetrable channels possess good retention behaviors).

Authors' response:

Thank you for your valuable comments. **We have added the paired-pulse facilitation (PPF) test data in Supplementary Fig. 19 and Supplementary Note 7 in the revised Supplementary Information.**

As you pointed out, the retention behavior of the device shows short-term potentiation (STP) performance, instead of long-term potentiation (LTP). We proposed to apply the intrinsically flexible hybrid PI-Al₂O₃ dielectric for the synaptic transistor to achieve STP synaptic performance and high flexibility simultaneously.

Changes: Yes

Location of Change in manuscript:

*Section: **Supplementary Information***

*Page: **Page 23, Supplementary Figure 19 and Supplementary Note 7 (green highlighted)***

*Section: **Homosynaptic characteristics of IFMRT***

*Page: **Page 11, Paragraph 1, Line 6 (green highlighted)***

3. In Figure 6e and g, the notation of ‘inhibitory mode’ and ‘excitatory mode’ is confusing. With PG = -0.5 V, the PSC is increasing, shouldn't this be the ‘excitatory mode’? Also, the modulation by PG is asymmetric for positive and negative TG pulses. This may limit the application of the device, as synaptic strengths relative to positive and negative presynaptic signals may differ substantially at a given PG voltage.

Authors' response:

Thank you for the valuable comments. We have double-checked and confirm that the notations of ‘inhibitory mode’ and ‘excitatory mode’ in Fig. 6e-g are correct following the traditional definition. When the presynaptic and postsynaptic pulses are in opposite directions, it is defined as the inhibitory mode. When the presynaptic and postsynaptic pulses are in identical directions, it is defined as the excitatory mode. The notations in Fig. 6e-g are consistent with this definition. **The definitions of ‘inhibitory mode’ and ‘excitatory mode’ are added in the last sentence of the first paragraph of the section “Artificial reconfigurable heterosynapse by IFMRT” as well as in Fig. 6b in the revised manuscript.**

While a symmetric modulation is expected for positive and negative TG pulses, asymmetric responses are frequently observed in practice, as reported in prior typical researches^{R3-R5}. Achieving synaptic response symmetry can indeed be addressed through waveform design, a well-established approach^{R6}. The primary focus of this work is to demonstrate *reconfigurable* synaptic responses—a novel contribution in this context. Given that waveform-based symmetry adjustment is a relatively mature technique, we have focused on the reconfigurability aspect.

References

- R3 Li, M. et al. Tailoring Neuroplasticity in a Ferroelectric - Gated Multi - Terminal Synaptic Transistor by Bi - Directional Modulation for Improved Pattern Edge Recognition. *Adv. Funct. Mater.* **33**, 2307986 (2023).
- R4 Wang, X. et al. Enhanced multiwavelength response of flexible synaptic transistors for human sunburned skin simulation and neuromorphic computation. *Adv. Mater.* **35**, 2303699 (2023).
- R5 Gao, C. et al. Feedforward Photoadaptive Organic Neuromorphic Transistor with Mixed-Weight

Plasticity for Augmenting Perception. *Adv. Func.l Mater.* **34**, 2313217 (2024).

R6 Park, J. et al. High-Performance Synapse Arrays for Neuromorphic Computing via Floating Gate-Engineered IGZO Synaptic Transistors. *Adv. Sci.* 2500568 (2025).

Changes: **Yes**

Location of Change in manuscript:

Section: **Artificial reconfigurable heterosynapse by IFMRT**

Page: **Page 12, Paragraph 1, Line 4 (green highlighted)**

4. In Figure 7, robotic motion control scenario is rather speculative. More complicated circuits and systems may be required to actuate the proposed signals into real robots hindering the applicability of the device. It is advised that a more realistic demonstration be provided.

Authors' response:

Thank you for the valuable comments. Implementing the functionality in Fig. 7c on an actual robot requires some peripheral circuits, such as sensors, oscillators, I-V TIA (Current-to-Voltage Transimpedance Amplifier) and comparator, as well as actuators serving as an artificial muscle^{R7}. These peripheral circuits are based on relatively mature PCB modules, as shown in the supplementary figures 9-11 of Ref. R7. In this scenario, the most critical component is the synaptic devices which can be implemented by our IFMRTs, while the peripheral circuits are less innovative and therefore saved originally.

Based on your advice, we supplemented the demonstration with commercially available collision sensors to make the demonstration more intuitive and realistic . The two collision sensors are connected to TG and BG, respectively, and they output a -0.5 V voltage when pressed. The practical test setup and the schematic diagram are shown in Fig. 7d-e in the revised manuscript. Corresponding to Fig. 7c, we controlled the waveforms of TG and BG as shown in Fig. 7f, and the measured PSC of the IFMRT is also presented. The relative magnitudes of the PSC in the four cases are consistent with those in the original results, as shown in Fig. R1.

Both Fig. 7 and the related descriptions have been updated in the revised manuscript.

Fig. R1 | The PSCs in the first version manuscript.

References

R7 Sun, F. et al. An artificial neuromorphic somatosensory system with spatio-temporal tactile perception and feedback functions[J]. *npj Flex. Electron.* **6**, 72(2022).

Changes: **Yes**

Location of Change in manuscript:

Section: **Dendrite integration by IFMRT for robotic decision and motion**

Page: **Page 15, Paragraph 1, Line 7; Page 15, Fig. 7d-f (green highlighted)**

5. In Figure 7e, the BG signal of (i) and (ii) are opposite signs. If a similar or identical thermal stimuli is applied shouldn't the BG (equivalent to spinal cord) signal be in the same sign?

Authors' response:

We sincerely appreciate the reviewer's careful reading and valuable feedback. Upon rechecking, we confirm that the BG signals in Fig. 7g (i) and Fig. 7g (ii) of the revised manuscript correctly share the same sign. However, we deeply regret that in the original submission, the TG and BG curves in Fig. 7e (ii) were inadvertently reversed during figure assembly, which may have caused confusion.

We have now corrected these curves in the revised manuscript and updated the original Fig. 7e to Fig. 7g, and carefully verified all figures to ensure accuracy. Thank you for bringing this to our attention, and we apologize for any inconvenience this oversight may have caused.

Changes: Yes

Location of Change in manuscript:

Section: Dendrite integration by IFMRT for robotic decision and motion

Page: Page 15, Fig. 7g

Reviewer #2:

Comments:

This manuscript introduces reconfigurable transistors with a dual-gate structure, where either the top or bottom gate can function as a modulation terminal to switch the conduction mode between n-type, p-type, or ambipolar configurations, while the other gate serves as a switch for the on/off state. These devices, termed intrinsically flexible multimode reconfigurable transistors (IFMRTs), demonstrate key-bit-selectable logic functions, providing potential solutions for hardware security. Additionally, the IFMRTs exhibit short-term synaptic properties, emphasizing their potential applications in neuromorphic computing. Their mechanical flexibility further enhances their suitability for wearable electronics and robotic systems.

It is interesting that a single device serving as a multimode reconfigurable transistor. However, certain aspects require further elaboration and detailed explanation regarding their applications are needed. Addressing these details would enhance the quality of the manuscript, making it more suitable for publication in Nature Communications. Here are specific comments and suggestions:

Authors' response:

We thank the reviewer for the positive summary for our work.

1. You mentioned "IFMRT array" in Fig. 1(e), but in the optical microscope image, the devices appear to be independent rather than interconnected as an array. Could you clarify whether these devices are electrically connected to form an actual array, or if "array" refers to their spatial arrangement?

Authors' response:

Thank you for raising this question. The array refers to their spatial arrangement. Each device is independent and not electrically connected.

We have modified the relevant description in the manuscript to avoid misleading.

Changes: Yes

Location of Change in manuscript:

Section: Device structure and characteristics of the IFMRT

Page: *Page 4, legend of Fig. 1e (green highlighted)*

2. You mentioned that top gating is more efficient due to its stronger coupling effect, while the weaker coupling of the bottom gate is attributed to damage in the underlying dielectric layer. However, a more detailed explanation of the relationship between coupling strength and dielectric damage would be beneficial.

Authors' response:

Thank you for the valuable comments. **We have added the detailed description to Supplementary Note 4 in the revised Supplementary Information and updated the related content in the revised manuscript.**

Changes: *Yes*

Location of Change in manuscript:

Section: **Device structure and characteristics of the IFMRT**

Page: *Page 5, Paragraph 2, Line 12 (green highlighted)*

Section: **Supplementary Information**

Page: *Page 8, Supplementary Note 4 (green highlighted)*

3. Additionally, it would be valuable to explore potential solutions for enhancing the bottom gate's coupling effect to achieve optimized performance, rather than simply selecting the better surface for mode switching. While the top gate (TG) is used as the polarity gate (PG) due to its superior coupling, the bottom gate (BG) still plays a crucial role as the switch for the on/off state. Given that the oxygen plasma etching induces asymmetric channel coupling, do you think adjusting the dielectric thickness could further enhance or mitigate this asymmetry? What strategies could be employed to optimize the bottom surface and improve its performance for better overall device efficiency?

Authors' response:

Thank you for the valuable comments. Yes, theoretical analysis suggest that scaling down the bottom dielectric thickness can increase the gate dielectric capacitance and decrease SS , bringing better gate control and mitigating this asymmetry. Besides, adjusting the oxygen plasma power and exposure duration could minimize interface state density while maintaining complete M-CNT etching, thereby improving SS under BG sweeping and enhancing overall device performance.

The potential solutions and strategies above have been added into the revised manuscript.

Changes: *Yes*

Location of Change in manuscript:

Section: **Device structure and characteristics of the IFMRT**

Page: *Page 5, Paragraph 2, Line 17 (green highlighted)*

4. How do the electrical properties (e.g., I_{on} , V_{th} , SS , mobility) vary with different channel sizes in IFMRTs? Could you provide statistical analysis across multiple devices to show uniformity and variability?

Authors' response:

Thank you for the valuable comments. We have provided the average I_{on} and statistical data of V_{th} , SS and mobility(μ) for devices with three different channel sizes, under four operation

modes: p-type modulated by BG, n-type modulated by BG, p-type modulated by TG, and n-type modulated by TG. The average I_{on} largely follows the expected W/L scaling relationship. The distributions of V_{th} , SS and μ are similar across devices with different channel dimensions.

We have added these statistic data as Supplementary Figs. 13-16 in the revised Supplementary Information.

Changes: Yes

Location of Change in manuscript:

*Section: **Supplementary Information***

*Page: **Page 17-20, Supplementary Figure 13-16***

*Section: **Device structure and characteristics of the IFMRT***

*Page: **Page 8, Paragraph 2, Line 17 (green highlighted)***

5. For the flexibility demonstration of the device, the manuscript states that the calculated bending stress is 1.6%. However, it would be beneficial to elaborate on the calculation or provide a more detailed explanation in the Supplementary Information, as flexible applications are a key aspect of this study.

Authors' response:

Thank you for the valuable comments. **We have added the calculation procedure as Supplementary Fig. 8 and Note 5 in the revised Supplementary Information according to your advice.**

Changes: Yes

Location of Change in manuscript:

*Section: **Supplementary Information***

*Page: **Page 12, Supplementary Figure 8 and Supplementary Note 5***

*Section: **Device structure and characteristics of the IFMRT***

*Page: **Page 8, Paragraph 2, Line 8 (green highlighted)***

6. The manuscript states that "micrometer-scale feature size also facilitates the uniformity of IFMRTs" and that "Supplementary Figs. 8 and 9 reflect a tight distribution, indicating the consistency of the performance." However, Figs. 8 and 9 only present data from 10 samples, which may not be a sufficiently large sample size to strongly support the claim of uniformity. Therefore, it is important to clarify whether the uniformity is primarily attributed to the intrinsic properties of the materials used in the device or if it is supported by the experimental data from these 10 samples.

- If the uniformity is driven by the general material properties, an appropriate reference should be cited to substantiate this claim.

- If the uniformity is based on the 10 experimental samples, an explanation should be provided to justify why this sample size is considered representative and how it aligns with standard practices for evaluating uniformity in this field.

Authors' response:

Thank you for helping us improving the accuracy of the writing. As you pointed out, 10 experimental samples are not enough to strongly support the uniformity of IFMRTs. We think the uniformity is driven by the general material properties, as the uniformity of single-gate transistors based on the CNTs was reported in our previous work^{R8}.

For enhanced accuracy in expression, we have revised the wording in the descriptions

for better accuracy for Fig. 3e-f and Supplementary Figs. 11 and 12.

References

R8 Huang, Q. et al. Intrinsically flexible all-carbon-nanotube electronics enabled by a hybrid organic – inorganic gate dielectric. *npj Flexible Electron.* **6**, 61(2022).

Changes: *Yes*

Location of Change in manuscript:

Section: **Supplementary Information**

Page: *Page 15, legend of Supplementary Figure 11; Page 16, legend of Supplementary Figure 12; (green highlighted)*

Section: **Device structure and characteristics of the IFMRT**

Page: *Page 8, Paragraph 2, Line 14; Page 7, legend of Fig.3 (green highlighted)*

7. In Figure 4, when Key = 1, the upper (lower) IFMRT has 0V (3V) applied to TG. Interpreting this using Figure 3, as seen in panels a and c, the upper IFMRT exhibits a V-curve behavior where the minimum current flows at approximately 1V, making it OFF, whereas the lower IFMRT appears to always conduct about 0.1 μA , remaining ON. In this case, the lower IFMRT is always pulling down the output to GND, while the upper IFMRT is expected to be in a competitive state where it turns on at both 0V and 3V, attempting to pull the output up to 3 V. Combining these pieces of information, short circuit conduction is expected to occur when V_{in} (V_{bg}) is 0 V and 3 V. The reason for turning off at 3V is presumed to be because the on-state resistance of the lower IFMRT is greater than that of the upper IFMRT. Short circuit conduction is known to cause critical energy consumption in integrated circuits, and I would like to ask how this issue can be resolved in this device.

Authors' response:

Thank you for your valuable comments. Since the IFMRTs in Fig. 3 a and c are used for mechanical bending tests, for this test, we did not choose the IFMRTs with the best electrical properties which is suitable for circuits. This issue can be addressed by adjusting the number of S-CNTs in the channel and then V_{th} to shift the whole curve left to decrease the current at “ $V_{\text{bgs}} = 0 \text{ V}$ ”.

We have added the resulted device I-V curves and the detailed descriptions of the solution to Supplementary Figure 17 and Supplementary Note 6 in the revised Supplementary Information. We have also updated the related content in the manuscript.

Changes: *Yes*

Location of Change in manuscript:

Section: **Supplementary Information**

Page: *Page 21, Supplementary Figure 17 and Supplementary Note 6 (green highlighted)*

Section: **Device structure and characteristics of the IFMRT**

Page: *Page 8, Paragraph 2, the last sentence (green highlighted)*

8. If the source and drain are metal and the device operates by changing the number of carriers in the channel, wouldn't a similar result be obtained with any semiconductor material if the same top-bottom sandwich gate structure is used instead of necessarily using a CNT channel? I would like to confirm whether this phenomenon can only be implemented with CNTs or if there is another reason why CNTs must be used.

Authors' response:

Thank you for your valuable comments. CNTs do not represent the sole semiconductor material for achieving reconfigurability. However, **realizing reconfigurable transistors requires semiconductor materials with ambipolar characteristics, as exemplified by two-dimensional materials, including black phosphorus (BP)^{R9}, WSe₂^{R10} and MoTe₂^{R10}, along with CNTs in this work.** The materials exhibiting only unipolar characteristics are fundamentally incapable of achieving reconfigurable transistor^{R10}. Furthermore, among the semiconductors with ambipolar properties above, **CNTs have better flexibility, compatible metal-semiconductor contacts, and better large-area integration potentials.**

References

- R9 Wu, P., Reis, D., Hu, X. S. & Appenzeller, J. Two-dimensional transistors with reconfigurable polarities for secure circuits. *Nat. Electron.* **4**, 45-53 (2021).
- R10 Sun, X. et al. Reconfigurable logic-in-memory architectures based on a two-dimensional van der Waals heterostructure device. *Nat. Electron.* **5**, 752–760 (2022).

Changes: No

9. I am having difficulty discerning the fundamental distinction between Fig. 6 and Fig.7. In both cases, it appears that the channel serves as a dendritic postsynaptic membrane. Could you clarify the key differences in their functional roles?

Authors' response:

Thank you for raising the confusion. Fig. 6 and Fig. 7 demonstrate two different application scenarios with different input schemes. Fig. 6 illustrates heterosynaptic plasticity, in which the presynaptic neuron is the dominant input while the modulatory neuron works as a modulator. Fig. 7 illustrates dendrite integration, in which both presynaptic membranes work as inputs equally.

We have added the description of the main difference in the revised manuscript. We have also modified the Fig.6a and Fig.7a to make it clearer.

Changes: Yes

Location of Change in manuscript:

Section: Dendrite integration by IFMRT for robotic decision and motion

Page: Page 15, Fig.7a; Page 14, Paragraph 2, Line 7 (green highlighted)

Section: Artificial reconfigurable heterosynapse by IFMRT

Page: Page 13, Fig.6a

10. It seems that ΔI was used to confirm STP characteristics. It would be good to add the V_{ds} conditions for reading I_{ds} in the main text. Also, adding the I_d - V_d (output curve) of the IFMRT would make the paper even better.

Authors' response:

Thank you for your valuable comments. **We have added V_{ds} in Fig. 5-7 of the revised manuscript, which is set to -0.1 V.**

The output characteristic curves (I_d - V_d) of p-type and n-type IFMRTs have been added as Supplementary Fig. 5 in the revised Supplementary Information.

Changes: Yes

Location of Change in manuscript:

Section: Homosynaptic characteristics of IFMRT

Page: Page 11, legend of Fig. 5c-f (green highlighted)

Section: Artificial reconfigurable heterosynapse by IFMRT

Page: Page 13, legend of Fig. 6d-e, Fig. 6h-j (green highlighted)

Section: Dendrite integration by IFMRT for robotic decision and motion

Page: Page 15, legend of Fig. 7f-g (green highlighted)

Section: Supplementary Information

Page: Page 22, legend of Supplementary Figure 18; Page 24-26, legend of Supplementary Figure 20, 21 and 22 (green highlighted); Page 9, Supplementary Figure 5

Section: Device structure and characteristics of the IFMRT

Page: Page 8, Paragraph 1, Line 2 (green highlighted)

11. The manuscript states that this design provides a solution for hardware security, saying that the transistor polarity remains hidden without the correct key. However, if a hypothetical opponent were able to map out the circuit layout, the key, which is a gating mechanism responsible for switching the transistor polarity, could also be exposed or accessed. Given this possibility, a more detailed explanation is needed to clarify how the polarity-gating effect contributes to hardware security. Elaborating on this aspect would enhance the reader's understanding and further support the novelty of the work, reinforcing its significance in the context of secure circuit design.

Authors' response:

Thank you for your valuable comments. **To facilitate reader comprehension, we have added a more detailed explanation into the revised manuscript based on your advice.**

Changes: Yes

Location of Change in manuscript:

Section: Reconfigurable polymorphic logic gates by IFMRT

Page: Page 9, Paragraph 1, Line 1 (green highlighted)

Section: References

Page: Page 20, Ref 43 (green highlighted)

12. Currently, the simulation seems to conclude with distinguishing binary inputs. In actual biological neurons, a neuron should be able to fire by receiving inputs from dozens of pre-neurons, and in the case of robotic movements, much more information may need to be processed beyond binary. Accordingly, if future work suggests ways to increase the number of input bits in IFMRT, it could make the paper more complete.

Authors' response:

Thank you for your valuable comments. Yes, increasing the number of inputs of the IFMRT would enable a more comprehensive simulation of dendritic structures of biological neurons and expand its applicability to more scenarios where robots require multiple inputs. **This can be achieved by increasing the number of TG and BG configurations in the IFMRT. Besides, cascading IFMRTs presents a viable solution.** These methods will be incorporated into our future research plan based on your advice.

Changes: No

Reviewer #3:

Comments:

The paper describes the implementation of intrinsically flexible multimode reconfigurable

transistors (IFMRTs) with dual-gate structure. Thanks to the modulation of the transistor modes, n- and p-types, as well as ambipolar, the IFMRTs are implemented for inverters and logic circuits realization, and as neuromorphic devices, achieving synaptic response. In addition, engineered solutions to tackle robotic motion and arm movement are reported. We demonstrate the solutions for the automatic obstacle avoidance and the coordinated movement control by low-level and high-level neural centers.

Authors' response:

We thank the reviewer for the summary of our work.

The following points should be clarified to edit the paper and improve the overall quality:

1. What is the state-of-the-art related to “reconfigurable transistors” and “flexible reconfigurable transistors”? This point plays a crucial role to understand how this manuscript can impact the scientific community;

Authors' response:

Thank you for your valuable comments. **We have added the comparison description into the Supplementary Note 1 in the revised Supplementary Information and the state-of-the-art have also been updated in the revised manuscript. The comparison table is added as Supplementary Table 1 in the revised Supplementary Information.**

Changes: Yes

Location of Change in manuscript:

*Section: **Methods, Device and circuit fabrication***

*Page: **Page 3, Paragraph 1, the last two sentences (green highlighted)***

*Section: **Supplementary Information***

*Page: **Page 2, Supplementary Table 1, Supplementary Note 1 (green highlighted)***

2. The device fabrication (page 3, paragraph “Methods” and Supplementary figure 1) is not described in details. In particular:

- No information is provided for the “M-CNT suspension”;
- No information is provided for the oxygen plasma etching (power, time, etc.);
- No information is provided for the “S-CNT suspension”;
- Which kind of patterning is employed for the Tog Gate electrode?
- What is the concentration for the phosphoric acid for the etching of BG and S/D pads?

Authors' response:

Thank you for your valuable comments. **All the above-mentioned details have been added to the Methods section of the revised manuscript based on your advice.**

In detail,

- The M-CNT suspension was purchased from Chinese Academy of Sciences Chengdu Organic Chemistry Co. Ltd., and was diluted from an original concentration of 2.0 wt.% to 0.5 wt.% for experimental use.
- The plasma treatment was performed using a gas mixture of nitrogen (3 L/min) and oxygen (5 cm³/min) at a power of 100 W. The treatment duration was 35 minutes for M-CNTs and 20 minutes for S-CNTs, respectively.
- The SCNT suspension consisted of 0.01 mg mL⁻¹ SCNT, which was prepared by ultrasonically 0.5 mg SCNT (NanoIntegris Inc., 99.9%) and 0.5 mg poly[(m-phenylenevinylene)-co-(2,5-dioctoxy-p-phenylenevinylene)] (PmPV) in 50 mL 1,2-

dichloroethane for 6 h.

- Consistent with BG patterning, TG electrodes were patterned by photolithography and oxygen plasma etching.
- The concentration for the phosphoric acid is 85%.

Changes: Yes

Location of Change in manuscript:

*Section: **Methods, Device and circuit fabrication***

Page: Page 17, Paragraph 3, Line 4; Page 18, Paragraph 1, Line 4 and Line 12 (green highlighted)

3. Can the authors explain the definition of “intrinsically flexible layer design for the dielectric”? This concept is quite unclear for the lack of proper explanation in the manuscript and for the lack of reference 41 in the “reference” paragraph;

Authors’ response:

Thank you for your valuable comments. The dielectric layer used in this work is a hybrid dielectric consisting of multi-layer polyimide and Al₂O₃ (hybrid PI-Al₂O₃) stacks. It is fabricated by a scalable vapor deposition technique consisting of molecular layer deposition (MLD) and atomic layer deposition (ALD), namely MLD/ALD. In this approach, MLD and ALD are alternatively executed to grow the polyimide and Al₂O₃ layers, respectively, with subnanometer thickness, and thus a uniform mixing of the polymer and oxide could be achieved at a nanometer level. Such a level of stacking could endow the merits of both constitutes to the dielectric, so that the afforded hybrid polyimide-Al₂O₃ dielectric has achieved intrinsically flexible mechanical performance and high-k dielectric performance simultaneously. That is why we call it “intrinsically flexible layer design for the dielectric”.

We have revised the descriptions of hybrid PI-Al₂O₃ dielectric in the manuscript to make it clearer. We have added the missing reference 41 in References section.

Changes: Yes

Location of Change in manuscript:

*Section: **References***

Page: Page 20, Ref 41 (green highlighted)

*Section: **Device structure and characteristics of the IFMRT***

Page: Page 4, Paragraph 1, Line 8 (green highlighted)

4. The authors claim that the IFMRT array shows “good transmittance”. Can they provide precise transmittance value in the visible spectrum (layer by layer and for the whole device stuck)? In addition, they should clarify how the Supplementary Figure 2 is acquired: what “without substrate” means? Was the sample with the array used for the transmittance measurement? Or was there a new sample with the unstructured layers deposited one on top of each other used to simulate the transmittance through the whole device stuck? This will give a better feeling to the reader on the transmittance of the devices;

Authors’ response:

Thank you for your valuable comments. **We have added precise transmittance values for each individual layer and the complete IFMRT array, measured at 500 nm, 700 nm, and 800 nm wavelengths, under both “with substrate” and “without substrate” conditions, as detailed in Supplementary Table 2.**

We have also added detailed descriptions in Supplementary Fig. 2 and Supplementary

note 2 in the revised Supplementary Information to make it more readable. Supplementary Fig. 2a-e presents the optical transmittance of each layer and the IFMRT array across the visible light spectrum. As illustrated in Supplementary Fig. 2f, the transmittance measurements were conducted using UV-2600 spectrophotometer from SHIMADZU CORPORATION, which employs a dual-fixture configuration: one for the reference and the other for the test sample. We used the sample with the IFMRT array for the transmittance measurement, not another new sample with the unstructured layers to simulate the whole device stack. The “with substrate” means that the transmittance was measured without any reference. The “without substrate” means that we used a blank substrate as reference and measured the substrate-normalized transmittance.

Changes: Yes

Location of Change in manuscript:

Section: Device structure and characteristics of the IFMRT

Page: Page 4, Paragraph 1, the second-to-last sentence (green highlighted)

Section: Supplementary Information

Page: Page 5-6, Supplementary Table 2, Supplementary Figure 2 and Supplementary Note 2

5. The authors claim “The ambipolarity of the S-CNT channel arises from the n-type doping of Al_2O_3 in the upper and lower dielectric layers.”. Can they provide scientific explanation of this doping? Have they performed any material analysis of the dielectric layer to support this statement?

Authors’ response:

Thank you for your valuable comments. **We have added the scientific explanation of the n-type doping in Supplementary Note 3 in the revised Supplementary Information.**

We have added the spectra by X-ray Photoelectron Spectroscopy (XPS) of the ALD deposited Al_2O_3 film to Supplementary Figure 3 in the revised Supplementary Information.

Changes: Yes

Location of Change in manuscript:

Section: Supplementary Information

Page: Page 7, Supplementary Figure 3 and Supplementary Note 3

Section: Device structure and characteristics of the IFMRT

Page: Page 4, Paragraph 2, Line 4 (green highlighted)

6. What is the leakage gate current in Figure 2b and 2c? These trends should be reported also in these plots for the sake of comprehensiveness (besides Supplementary Information);

Authors’ response:

Thanks for the advice. **We have supplemented the gate leakage current in Fig.2j and Fig.2k corresponding to Fig.2b and Fig.2c.** The gate leakage current remains below 1 nA regardless of whether TG or BG acting as PG.

Changes: Yes

Location of Change in manuscript:

Section: Device structure and characteristics of the IFMRT

Page: Page 6, Fig.2j-k; Page 7, Paragraph 1, Line 19 (green highlighted)

7. Can the authors provide scientific support to “During the oxygen plasma etching the S/D electrodes, the upper surface of the bottom dielectric layer suffers minor damage, leading to the weaker coupling capability of BG.” ? Was there any damage visible on the dielectric surface? More details should be provided;

Authors’ response:

Thank you for your valuable comments. **We have added the scientific support into Note 4 in the revised Supplementary Information. We have also updated the related discussion in the manuscript to make it clearer.**

Under microscopic examination, no visible damage was observed on the dielectric surface, as shown in Fig. R2 below.

Fig.R2 | Optical microscope image of IFMRTs after sequentially fabricating BG, the first dielectric and S/D electrodes. The dashed-line area marks the first dielectric layer of the channel region, showing no visible surface damage.

Changes: Yes

Location of Change in manuscript:

Section: Device structure and characteristics of the IFMRT

Page: Page 5, Paragraph 2, Line 12 (green highlighted)

Section: Supplementary Information

Page: Page 8, Supplementary Note 4 (green highlighted)

8. In the Reference paragraph, there are 40 references, whereas, the authors use 42 references in the manuscript;

Authors’ response:

Thank you for pointing this out. **We have added the two missing references.** Ref 41 and Ref 42 in the original version manuscript has been updated to Ref 41 and Ref 44 in the revised manuscript.

Changes: Yes

Location of Change in manuscript:

Section: References

Page: Page 20, Ref 41, Ref 44 (green highlighted)

Point-by-point responses to reviewer comments

Reviewer #1:

1. While the authors have provided a response, several important concerns remain regarding the practical implementation of the proposed device and system. Firstly, the device is intended to function both as a conventional logic circuit and as a synaptic transistor. However, synaptic transistors typically rely on the slow relaxation dynamics of the PI-Al₂O₃ dielectric, which inherently limits switching speeds. This trade-off raises concerns about the suitability of the device for conventional high-speed binary logic operations, potentially restricting its applicability in real electronic systems.

Authors' response:

We thank the reviewer for this insightful comment regarding the critical trade-off between hysteresis for synaptic functions and switching speed for logic operations. We agree that this is a fundamental consideration for the practical implementation of reconfigurable devices.

The core of our approach is that the hysteresis and switching speed of the IFMRT are not fixed but can be designed and tuned for the specific target application.

- 1) For high-speed logic circuits, we can fabricate the PI-Al₂O₃ dielectric to minimize the generation of polar groups (NH₃⁺ and COO⁻), thereby effectively eliminating hysteresis and enabling faster switching^[Ref.1]. Our previous work^[Ref.2] demonstrated this capability by achieving a ring oscillator with a frequency of 1.75 kHz using single-gate transistors. This kHz-level operating speed is indeed suitable for applications like flexible displays and biomedical signal sensing.
- 2) For synaptic transistors, we intentionally engineer the dielectric to have a significant hysteresis window, which is essential for emulating synaptic plasticity.

This tunability is achieved by modulating the molecular layer deposition (MLD) and atomic layer deposition (ALD) cycles during the fabrication of the PI-Al₂O₃ dielectric^[Ref.1], a method we established in Fig. 2 of [Ref.1], as below.

Figure 2 of [Ref.1]. Schematic illustration of the supercycle parameters in the HPA layer deposition process. Dark blue represents PMDA vapor pulse, white represents N₂ purge, yellow represents EDA vapor pulse, light blue represents TMA vapor pulse, and green represents deionized water vapor pulse. (a) A supercycle consisting of 2 MLD cycles and 2 ALD cycles. (b) A supercycle consisting of 6 MLD cycles and 6 ALD cycles. Transfer characteristic curves of transistors fabricated with HPA dielectric layers deposited under different process parameters. The drain-source bias (V_{DS}) is 10 mV. The graph also shows the gate leakage current. (c) 30 nm HPA dielectric layer deposited using 2 MLD + 2 ALD cycles. (d) 30 nm HPA dielectric layer deposited using 6 MLD + 6 ALD cycles. Schematic diagram of the microstructure of the HPA layer: (e) HPA layer obtained by the alternating deposition of 2 MLD cycles and 2 ALD cycles; (f) HPA layer obtained by the alternating deposition of 6 MLD cycles and 6 ALD cycles.

cycles and 6 ALD cycles.^[Ref.1]

In this submitted work, we have realized the intrinsically flexible reconfigurable transistors and validated their functionality in both integrated circuits and neuromorphic devices to demonstrate their versatility. The reviewer's insightful comment highlights the importance of explicitly discussing the modulation of hysteresis, which is also an important mechanism enhancing this reconfigurability besides the dual-gate architecture. **Accordingly, we have revised the manuscript to clearly articulate that the device properties are application-specific and can be modulated via the fabrication process. This along with the comparison in Supplementary Table 1 and Note 1, further shows the advantages of the work.**

References

Ref. 1 (Reference [42] in the manuscript) Zhu, J. et al. Ultra-flexible high-linearity silicon nanomembrane synaptic transistor array. *Adv. Mater.* 2413404 (2025).

Ref. 2 (Reference [41] in the manuscript) Huang, Q. et al. Intrinsically flexible all-carbon-nanotube electronics enabled by a hybrid organic–inorganic gate dielectric. *npj Flexible Electron.* 6, 61 (2022).

Changes: Yes

Location of Change in manuscript:

*Section: **Device structure and characteristics of the IFMRT***

*Page: **Page 8, Paragraph 1, Line 4 (green highlighted)***

2. Furthermore, in Figure 7, the demonstration of robotic control appears limited in scope, as the robotic movements are predetermined. Real-world robotic applications typically require more complex and adaptive device behaviors. For reference, more advanced demonstrations can be found in works such as Nature Communications 14, 5 (2023) and Science Advances 8, eabo3326 (2022), where device functionalities are more closely aligned with the demands of dynamic and responsive robotic systems.

Authors' response:

We thank the reviewer for their insightful comment regarding the scope of our robotic demonstration and for highlighting the advanced system-level work of Roe et al. and Kim et al. We agree that these studies present significant applications in adaptive robotic control, implemented on conventional, rigid platforms with components of fixed functionality. Our work, by contrast, addresses a distinct and foundational device challenge. The Intrinsically Flexible Multimode Reconfigurable Transistors (IFMRTs) provide a paradigm shift by introducing two key material- and device-level innovations:

1. **Intrinsic Flexibility:** This enables the direct embodiment of neuromorphic computation within non-planar robotic structures, moving beyond centralized processing toward distributed, morphological computation.
2. **Single-Device Multimode Reconfigurable Functionality:** The dynamic reconfigurability between synaptic, dendritic, and neuronal modes within a single device provides a level of functional adaptability.

The robotic function integration in this work only serves as a proof of concept for these core device properties. As illustrated in Figure 7, a simplified experimental setup was deliberately employed to demonstrate the IFMRT can be integrated into a physical system for basic robotic

decision-making and arm control. This approach, utilizing predetermined movements, isolates interference from complex environmental variables, thereby enabling a clearer establishment of causality between the device's output and the resulting robotic behavior. **It validates the fundamental operational principles that enable the pursuit of more complex and adaptive systems. Furthermore, for the researches the reviewer shared, the IFMRTs hold the potential to enhance the robotic systems by enabling higher-level integration and multifunctional capabilities within a simplified form.**

In brief, the IFMRTs are not an end-point application but an enabling technology that provides the device foundation to surpass the current limitations in neuromorphic robotics, paving the way for future systems with higher levels of integration and adaptability. The reviewer's suggestion to pursue full system integration is an important direction for future work.

We have added a discussion to the "Dendrite integration by IFMRT for robotic decision and motion" section addressing the future development to integrate the IFMRTs for a more complex and adaptive device behaviors. Relevant references have been included.

Changes: Yes

Location of Change in manuscript:

*Section: **Dendrite integration by IFMRT for robotic decision and motion***

Page: Page 17, Paragraph 2, Line 3 (green highlighted)

*Section: **References***

Page: Page 20, Ref 45 and Ref 46 (green highlighted)

3. In the context of this work, the advantages of synaptic plasticity appear to be underutilized or unnecessary for the demonstrated application. Therefore, the practical value of employing a multifunctional device in this manner may be limited.

Authors' response:

We appreciate your insightful comments. The value of synaptic plasticity in the multifunctional device is evidenced by its successful application in two distinct scenarios with different input schemes.

Specifically, Figure 6 illustrates heterosynaptic plasticity, featuring a dominant presynaptic input modulated by a secondary neuron. Figure 7, on the other hand, demonstrates dendritic integration for robotics, where two presynaptic inputs have equal weight.

This ability to reconfigure a single device for different functions underscores the critical role and versatility of synaptic plasticity.

Reviewer #2:

I conducted a thorough examination of the revised manuscript and SI. I am satisfied that the response letter adequately addresses all of the concerns I raised. and the manuscript and SI are well revised. I was grateful for the author's dedication to improving the manuscript's quality.

Authors' response:

We thank the reviewer for their positive assessment of our revised manuscript.

Reviewer #4:

This manuscript reports the development of an intrinsically flexible multimode reconfigurable transistor (IFMRT), integrating dual-gate control, ambipolar operation, and dendrite-inspired

neuromorphic functions. The authors demonstrate polymorphic logic gates, reconfigurable synaptic plasticity (including artificial heterosynapse behavior), and dendritic integration for robot decision-making and motor control, all using a single device architecture. The concept is novel and well-motivated, and the experimental demonstrations are convincing and multidisciplinary in scope, spanning circuit applications, synaptic behavior emulation, and robot control.

The manuscript is generally well written and logically structured. I believe this work has strong potential for publication. However, a few important clarifications and additions are required before I can recommend acceptance.

Authors' response:

We thank the reviewer for the positive summary of our work.

1. Transistor Operation Regime and Output Characteristics

All transistor characterizations appear to be performed in the linear regime (e.g., $V_{ds} = -0.5$ V), and no data are shown for the saturation regime (e.g., $V_{ds} = -3$ V), which is essential for digital logic and analog gain evaluation. It would strengthen the manuscript to include output characteristics in the saturation region and to discuss saturation current, output resistance, or intrinsic gain if available.

Authors' response:

Thank you for your valuable comments. **We have provided output characteristics in Supplementary Figure 5. We have added the calculations of saturation current, output resistance and intrinsic gain in Supplementary Note 5.**

Changes: Yes

Location of Change in manuscript:

Section: Device structure and characteristics of the IFMRT

Page: Page 8, Paragraph 1, Line 2 (green highlighted)

Section: Supplementary Information

Page: Page 9, Supplementary Note 5 (green highlighted)

2. Mobility Asymmetry and Circuit-Level Implications

Supplementary Figure 11 shows that the extracted field-effect mobility differs significantly between p-type ($7\text{--}14$ cm²/Vs) and n-type ($2.5\text{--}7.5$ cm²/Vs) modes when TG acts as the programming gate, and this difference becomes even larger when BG is used as the programming gate.

1) Could this mobility asymmetry be responsible for the observed shift of the inverter switching point toward the positive side of $V_{dd}/2$ in Fig. 4b?

Authors' response:

Thank you for raising this question. Yes. We also think that the mobility asymmetry induces the positive shift of the inverter switching point.

2) In conventional complementary logic circuits, mobility mismatch is typically compensated by adjusting the W/L ratio of p- and n-type transistors. However, the main advantage of IFMRTs lies in their reconfigurability — the ability to define p- or n-type behavior after fabrication. If W/L must be predetermined to account for this asymmetry, does this not undermine that advantage?

Authors' response:

Thank you for raising this question. Yes, if effective mobility mismatch exists in logic circuits, it would undermine this advantage. However, the effective mobility mismatch is mainly induced by the difference between barrier heights at the metal-semiconductor contacts for electron and hole. Addressing this through precise Fermi level engineering of the contact electrodes is indeed a critical and promising avenue for optimizing device performance. We consider this a key focus of our ongoing and future work, as it requires a dedicated effort to meticulously tailor the metal-CNT interface. **We have added a discussion to highlight this important point and the potential solutions in the revised Supplementary as Note 8.**

Changes: Yes

Location of Change in manuscript:

*Section: **Device structure and characteristics of the IFMRT***

*Page: **Page 8, Paragraph 2, Line 17 (green highlighted)***

*Section: **Supplementary Information***

*Page: **Page 16, Supplementary Note 8 (green highlighted)***

3) In this context, do the authors consider the observed mobility imbalance and resulting shift in logic thresholds to be within acceptable margins for future high-density integration? Additionally, was the inverter characteristic in Fig. 4b obtained from a single-sweep measurement or during a bidirectional V_{in} sweep? Is there any observable hysteresis?

Authors' response:

Thank you for raising these questions. The mobility imbalance leads to a shift in the logic threshold voltage, which primarily degrades the noise margin of the inverter. We extracted the noise margin values based on the transfer curve shown in Fig. 4b. As shown in Fig. R1, the high noise margin is 0.53 V and the low noise margin is 0.5 V, about $V_{DD}/6$. For reference, according to "CMOS VLSI Design: A Circuits and Systems Perspective" by Weste and Harris ^[Ref.3], typical noise margins are around $V_{DD}/3$ in older process technologies, and reduce to $V_{DD}/4$ or less in nanometer-scale nodes. This shows that the noise margins of the inverter are relatively small. Future work on high-density integration will focus on improving the n-type and p-type mobility balance by contact engineering to shift the switching point closer to $V_{DD}/2$, thereby increasing the noise margins.

The inverter characteristic in Fig. 4b was obtained using a single-sweep measurement. **We have also supplemented the hysteresis of the inverter in Supplementary Fig. 18, which exhibits a clockwise hysteresis behavior.**

References

Ref.3 Weste, N. H. E. & Harris, D. M. CMOS VLSI Design: A Circuits and Systems Perspective (4th edn, Pearson, 2011).

Fig. R1 | The extracted noise margin of the inverter in Fig. 4b.

Changes: Yes

Location of Change in manuscript:

Section: Reconfigurable polymorphic logic gates by IFMRT

Page: Page 10, Paragraph 1, Line 11 (green highlighted)

Section: Supplementary Information

Page: Page 23, Supplementary Figure 18 (green highlighted)

3. Mobility Extraction Methodology

While mobility values are provided in Supplementary Figure 11, the extraction method is not described in the manuscript or supplementary information. Please include a brief explanation of the calculation procedure (e.g., equations used, assumptions about gate capacitance or threshold extraction), even if mobility is extracted from the linear regime.

Authors' response:

Thank you for your valuable comments. We have added the explanation of the calculation procedure in Supplementary Note 7 in the revised supplementary information.

Changes: Yes

Location of Change in manuscript:

Section: Device structure and characteristics of the IFMRT

Page: Page 8, Paragraph 2, Line 17 (green highlighted)

Section: Supplementary Information

Page: Page 16, Supplementary Note 7 (green highlighted)

4. Device Geometry and Scaling Discussion

The channel length variation ($L = 20, 50, 100 \mu\text{m}$) is mentioned in the Supplementary Information, but the channel width (W) and W/L ratios used in the main device demonstrations are not clearly stated.

Please specify the W/L ratios for devices used in logic circuits and neuromorphic applications.

Authors' response:

Thank you for your valuable comments. We have added the values of W and L in the revised manuscript and revised supplementary information.

Changes: Yes

Location of Change in manuscript:

Section: Device structure and characteristics of the IFMRT

Page: Page 6, legend of Fig. 2; Page 7, legend of Fig. 3 (green highlighted)

Section: Reconfigurable polymorphic logic gates by IFMRT

Page: Page 9, legend of Fig. 4 (green highlighted)

*Section: **Homosynaptic characteristics of IFMRT***

Page: Page 11, legend of Fig. 5 (green highlighted)

*Section: **Artificial reconfigurable heterosynapse by IFMRT***

Page: Page 13, legend of Fig. 6 (green highlighted)

*Section: **Dendrite integration by IFMRT for robotic decision and motion***

Page: Page 15, legend of Fig. 7 (green highlighted)

*Section: **Supplementary Information***

Page: Page 8, legend of Supplementary Figure 4; Page 9, legend of Supplementary Figure 5; Page 22, legend of Supplementary Figure 17; Page 24, legend of Supplementary Figure 19; Page 25, legend of Supplementary Figure 20; Page 26, legend of Supplementary Figure 21; Page 27, legend of Supplementary Figure 22; Page 28, legend of Supplementary Figure 23; (green highlighted)

Recommendation: Minor Revision

The manuscript introduces a promising class of flexible, reconfigurable transistors and demonstrates their use in logic and neuromorphic applications, including robot control. However, critical points related to mobility asymmetry, scalability, and logic-level design must be addressed to fully support the claims and enable broader applicability. I encourage the authors to consider the above comments carefully to strengthen the manuscript.

Authors' response:

We thank the reviewer for their positive assessment and valuable comments for improving our revised manuscript.

Point-by-point responses to reviewer comments

Reviewer #1:

The authors fully addressed the comments raised by all the reviewers.

So, this paper can be published in Nature Communications without further revisions.

Authors' response:

We sincerely appreciate your positive feedback on our work. We are pleased that our responses have fully addressed all the issues you raised. We are delighted and honored by your decision to accept our manuscript.

Reviewer #4:

We appreciate the authors' dedicated efforts and the comprehensive revisions made to the manuscript and the Supplementary Information. All key concerns raised during the review process have been addressed in a scientifically rigorous and transparent manner.

Specifically:

1. Transistor Characteristics: The addition of output characteristics, saturation current, output resistance, and intrinsic gain in Supplementary Note 5 and Figure 5 is highly valuable and strengthens the evaluation of both the digital and analog potential of the IFMRTs.

2. Mobility Asymmetry and Scalability: The transparent discussion of the mobility mismatch between p-type and n-type operation, the resulting shift in the logic threshold, and the current noise margin ($\approx V_{DD}/6$) is commendable. The proposal of contact engineering (Supplementary Note 8) as the critical path for future optimization is a crucial and well-articulated strategy for high-density integration.

3. Technical Details: The inclusion of the mobility extraction methodology (Supplementary Note 7), as well as the specific W and L values in the manuscript, significantly improves the reproducibility and clarity of the work.

4. Robotic Application: The clarification that the robotic demonstration serves as a proof-of-concept for the device's foundational reconfigurability and intrinsic flexibility is satisfactory, aligning the scope of the work with its core device innovation.

The manuscript presents a highly innovative and important advancement in the field of intrinsically flexible, multifunctional electronics. The demonstrated integration of reconfigurable logic and neuromorphic functions in a single flexible platform is novel and holds strong potential for future soft robotics and wearable technologies.

The manuscript is now recommended for acceptance.

Authors' response:

Thank you for the approving and detailed comments to our revision. We are delighted that our revisions have thoroughly addressed all the points you raised. Your insightful comments have significantly improved the quality of our manuscript. It is a great honor to receive your decision to accept our manuscript.

This manuscript introduces reconfigurable transistors with a dual-gate structure, where either the top or bottom gate can function as a modulation terminal to switch the conduction mode between n-type, p-type, or ambipolar configurations, while the other gate serves as a switch for the on/off state. These devices, termed intrinsically flexible multimode reconfigurable transistors (IFMRTs), demonstrate key-bit-selectable logic functions, providing potential solutions for hardware security. Additionally, the IFMRTs exhibit short-term synaptic properties, emphasizing their potential applications in neuromorphic computing. Their mechanical flexibility further enhances their suitability for wearable electronics and robotic systems.

It is interesting that a single device serving as a multimode reconfigurable transistor. However, certain aspects require further elaboration and detailed explanation regarding their applications are needed. Addressing these details would enhance the quality of the manuscript, making it more suitable for publication in Nature Communications. Here are specific comments and suggestions:

- (1) You mentioned "IFMRT array" in Fig. 1(e), but in the optical microscope image, the devices appear to be independent rather than interconnected as an array. Could you clarify whether these devices are electrically connected to form an actual array, or if "array" refers to their spatial arrangement?
- (2) You mentioned that top gating is more efficient due to its stronger coupling effect, while the weaker coupling of the bottom gate is attributed to damage in the underlying dielectric layer. However, a more detailed explanation of the relationship between coupling strength and dielectric damage would be beneficial.
- (3) Additionally, it would be valuable to explore potential solutions for enhancing the bottom gate's coupling effect to achieve optimized performance, rather than simply selecting the better surface for mode switching. While the top gate (TG) is used as the polarity gate (PG) due to its superior coupling, the bottom gate (BG) still plays a crucial role as the switch for the on/off state. Given that the oxygen plasma etching induces asymmetric channel coupling, do you think adjusting the dielectric thickness could further enhance or mitigate this asymmetry? What strategies could be employed to optimize the bottom surface and improve its performance for better overall device efficiency?
- (4) How do the electrical properties (e.g., Ion, Vth, SS, mobility) vary with different channel sizes in IFMRTs? Could you provide statistical analysis across multiple devices to show uniformity and variability?

- (5) For the flexibility demonstration of the device, the manuscript states that the calculated bending stress is 1.6%. However, it would be beneficial to elaborate on the calculation or provide a more detailed explanation in the Supplementary Information, as flexible applications are a key aspect of this study.
- (6) The manuscript states that "micrometer-scale feature size also facilitates the uniformity of IFMRTs" and that "Supplementary Figs. 8 and 9 reflect a tight distribution, indicating the consistency of the performance." However, Figs. 8 and 9 only present data from 10 samples, which may not be a sufficiently large sample size to strongly support the claim of uniformity. Therefore, it is important to clarify whether the uniformity is primarily attributed to the intrinsic properties of the materials used in the device or if it is supported by the experimental data from these 10 samples.
- If the uniformity is driven by the general material properties, an appropriate reference should be cited to substantiate this claim.
 - If the uniformity is based on the 10 experimental samples, an explanation should be provided to justify why this sample size is considered representative and how it aligns with standard practices for evaluating uniformity in this field.
- (7) In Figure 4, when Key = 1, the upper (lower) IFMRT has 0V (3V) applied to TG. Interpreting this using Figure 3, as seen in panels a and c, the upper IFMRT exhibits a V-curve behavior where the minimum current flows at approximately 1V, making it OFF, whereas the lower IFMRT appears to always conduct about 0.1 μ A, remaining ON. In this case, the lower IFMRT is always pulling down the output to GND, while the upper IFMRT is expected to be in a competitive state where it turns on at both 0V and 3V, attempting to pull the output up to 3 V. Combining these pieces of information, short circuit conduction is expected to occur when V_{in} (V_{bg}) is 0 V and 3 V. The reason for turning off at 3V is presumed to be because the on-state resistance of the lower IFMRT is greater than that of the upper IFMRT. Short circuit conduction is known to cause critical energy consumption in integrated circuits, and I would like to ask how this issue can be resolved in this device.
- (8) If the source and drain are metal and the device operates by changing the number of carriers in the channel, wouldn't a similar result be obtained with any semiconductor material if the same top-bottom sandwich gate structure is used instead of necessarily using a CNT channel? I would like to confirm whether this phenomenon can only be implemented with CNTs or if there is another reason why CNTs must be used.

- (9) I am having difficulty discerning the fundamental distinction between Fig. 6 and Fig. 7. In both cases, it appears that the channel serves as a dendritic postsynaptic membrane. Could you clarify the key differences in their functional roles?
- (10) It seems that ΔI was used to confirm STP characteristics. It would be good to add the V_{ds} conditions for reading I_{ds} in the main text. Also, adding the I_d - V_d (output curve) of the IFMRT would make the paper even better.
- (11) The manuscript states that this design provides a solution for hardware security, saying that the transistor polarity remains hidden without the correct key. However, if a hypothetical opponent were able to map out the circuit layout, the key, which is a gating mechanism responsible for switching the transistor polarity, could also be exposed or accessed. Given this possibility, a more detailed explanation is needed to clarify how the polarity-gating effect contributes to hardware security. Elaborating on this aspect would enhance the reader's understanding and further support the novelty of the work, reinforcing its significance in the context of secure circuit design.
- (12) Currently, the simulation seems to conclude with distinguishing binary inputs. In actual biological neurons, a neuron should be able to fire by receiving inputs from dozens of pre-neurons, and in the case of robotic movements, much more information may need to be processed beyond binary. Accordingly, if future work suggests ways to increase the number of input bits in IFMRT, it could make the paper more complete.